# RA-TTA: Retrieval-Augmented Test-Time Adaptation for Vision-Language Models

**Youngjun Lee[1], Doyoung Kim[1], Junhyeok Kang[2], Jihwan Bang[1], Hwanjun Song[1], Jae-Gil Lee[1,*]**
[1] KAIST, [2] LG AI Research
{youngjun.lee, doyo09, jihwan.bang, songhwanjun, jaegil}@kaist.ac.kr
junhyeok.kang@lgresearch.ai

## ABSTRACT

Vision-language models (VLMs) are known to be susceptible to distribution shifts between pre-training data and test data, and test-time adaptation (TTA) methods for VLMs have been proposed to mitigate the detrimental impact of the distribution shifts. However, the existing methods solely rely on the internal knowledge encoded within the model parameters, which are constrained to pre-training data. To complement the limitation of the internal knowledge, we propose *Retrieval-Augmented-TTA (RA-TTA)* for adapting VLMs to test distribution using *external* knowledge obtained from a web-scale image database. By fully exploiting the bi-modality of VLMs, RA-TTA *adaptively* retrieves proper external images for each test image to refine VLMs' predictions using the retrieved external images, where fine-grained *text descriptions* are leveraged to extend the granularity of external knowledge. Extensive experiments on 17 datasets demonstrate that the proposed RA-TTA outperforms the state-of-the-art methods by 3.01–9.63% on average.

## 1 INTRODUCTION

In recent years, vision-language models (VLMs) pre-trained on large corpora of image-text pairs have garnered significant attention (Radford et al., 2021; Jia et al., 2021; Li et al., 2022; Sun et al., 2023; Zhang et al., 2024a). When transferring the pre-trained knowledge of VLMs at test time, distribution shifts between pre-training data and test data deteriorate the zero-shot transferability of VLMs (Bommasani et al., 2021; Nguyen et al., 2022; Fang et al., 2022; Santurkar et al., 2023). Thus, a number of *test time adaptation (TTA)* methods for VLMs (Shu et al., 2022; Feng et al., 2023; Ma et al., 2023; Zhao et al., 2024b; Karmanov et al., 2024; Zhang et al., 2024b) have been proposed to mitigate the detrimental impact of the distribution shifts. For adapting to an unlabeled test image, the existing methods typically rely on the outputs of VLMs on the test image, which are determined by the pre-trained knowledge encoded within the model parameters. However, this internal knowledge learned from the pre-training data may be insufficient to address unseen test data potentially deviated by distribution shifts (Agarwal et al., 2021; Menon et al., 2024; Parashar et al., 2024).

Due to the difficulty of updating pre-trained models frequently, retrieval-augmented generation (RAG) (Gao et al., 2023; Zhao et al., 2024a; Fan et al., 2024) is proposed in language domains, and it incorporates *external knowledge* from a document database into a query to complement new or focused knowledge absent in the pre-trained knowledge. Thus, in accordance with the philosophy of using external knowledge, we propose a *retrieval-augmented* approach for TTA with VLMs and refer to this approach as *retrieval-augmented-TTA (RA-TTA)*. Figure 1(a) shows the overall procedure of RA-TTA: (i) a VLM is requested to provide an answer in response to a test image; (ii) the test image is queried against a web-scale *image* database to retrieve relevant and useful images for the purpose of TTA; (iii) the final answer is adjusted on the fly using both the answer from the VLM and the retrieved external images. The image database usually contains only the images without labels and captions (Caron et al., 2019; Tian et al., 2021; Goyal et al., 2021).

It is evident that the external images offer insights into a test image only when properly retrieved. For example, let's consider a task that recognizes a vehicle type (e.g., CX-9) from an image. If a test image depicts the front and side of a vehicle while obscuring the rear, the external images with

---

*Corresponding author.

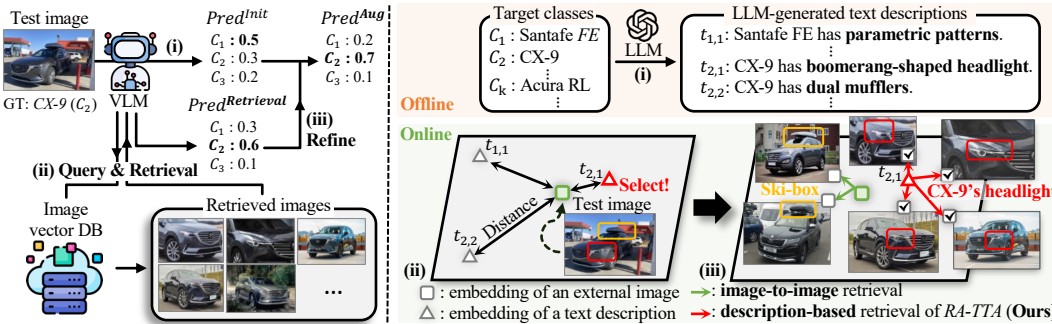

(a) Overview of the proposed RA-TTA.  (b) Overview of the description-based retrieval.

Figure 1: **Key idea of RA-TTA**: (a) shows the overall procedure of RA-TTA; (b) elaborates on the description-based retrieval approach, where a VLM embeds a test image into the embedding space and identifies its pivotal features (text descriptions), e.g., boomerang-shaped headlight of CX-9, and the neighboring images to the identified descriptions in the embedding space are retrieved.

*pivotal* features such as the headlight must be helpful, whereas those with non-visible features such as the taillight or irrelevant features such as a ski-box would not aid in the recognition. This objective aligns with RAG for retrieving documents that contain critical information relevant to a given query. However, naive image-to-image similarity search (Iscen et al., 2023) often fails to retrieve the images with the pivotal features. The primary reason is that an image, similar to a lengthy document, often encompasses multiple *diverse* semantics, e.g., a vehicle image with a headlight and a ski-box in Figure 1(b). That is, images may be quite intricate to be encapsulated by single embeddings. Recall that, in RAG, a document is divided into multiple chunks, with each chunk intended to contain a specific piece of information (Lewis et al., 2020; Wang et al., 2024a). This chunking enables the capture of more relevant and coherent information for a query. On the other hand, because precise image segmentation is very costly and challenging, applying image segmentation to achieve an effect analogous to document chunking for images is infeasible.

In this paper, to retrieve proper external images and ultimately improve the zero-shot transferability of a VLM at test time, we propose a *description-based retrieval* approach that fully leverages the bi-modality of VLMs. Figure 1(b) elaborates on its overall procedure: (i) *Multiple* (e.g., 20) *fine-grained* visual features for each target class are extracted in the form of text descriptions, such as "Mazda CX-9 has a boomerang-shaped headlight," by LLMs. These text descriptions are extracted offline prior to test time. (ii) For each test image given online, text descriptions relevant to a test image are selected by image-to-text similarity search on the embedding space of a VLM, which we regard as the pivotal features of the test image. (iii) External images aligned with these selected text descriptions are retrieved by text-to-image similarity search on the same embedding space. We contend that *image retrieval through text descriptions* is comparable to document chunking, as it divides images into multiple semantic chunks for effective retrieval. In Figure 1(b), due to the text descriptions on the headlight of CX-9, its headlights are desirably dominant in the retrieved images. In contrast, arbitrary SUV images with a ski-box could be retrieved if a naive image-to-image similarity search were employed. Moreover, we also propose a *description-based adaptation* approach that exploits the selected text descriptions for the adaptation with the retrieved images.

Overall, the proposed RA-TTA equipped with the description-based retrieval and adaptation significantly outperforms the state-of-the-art TTA methods on 17 datasets, including the standard transfer learning and natural distribution shift benchmarks. According to our extensive evaluation, the impact of external images was higher when the strength of distribution shifts became higher, and the advantages of the description-based retrieval became more pronounced when test data exhibited more complex semantics (see § 4.2 and § 4.6).

## 2 RELATED WORK AND PRELIMINARY

### 2.1 VISION-LANGUAGE MODELS

Vision-language models (VLMs), such as CLIP (Radford et al., 2021), ALIGN (Jia et al., 2021), and LongCLIP (Zhang et al., 2024a), are pre-trained to align images with corresponding text descriptions, thereby understanding the relationships between arbitrary image-text pairs. These

models have demonstrated excellent zero-shot transferability for image classification tasks, where classes are represented by text prompts, and images are classified based on the similarity to these prompts (Radford et al., 2021). Previous approaches employ *coarse* text prompts, such as "a photo of Chevrolet Impala," which do not fully utilize the contextual richness of language. Recent advancements shift towards incorporating *fine-grained* visual details in prompts, such as "Chevrolet Impala has sharp and muscular styling," to enhance representational capacity. For example, VisDesc (Menon & Vondrick, 2022) and CuPL (Pratt et al., 2023) employ class-specific, LLM-generated text descriptions as the prompts for VLMs. WaffleCLIP (Roth et al., 2023) proposes a novel prompting technique for VLMs to improve the utility of the text descriptions by analyzing the effect of LLM-generated text descriptions. Following these studies, our research further explores the utility of fine-grained text descriptions to improve VLM efficacy.

## 2.2 Test-Time Adaptation for Vision-Language Models

When transferring the zero-shot capabilities of VLMs, the distribution shift between pre-training data and test data is the main obstacle. Test-time adaptation (TTA) methods for VLMs have been proposed to adapt VLMs to the distribution shift. These methods adapt to an input test image on the fly without any training requirements. TPT (Shu et al., 2022) uses data augmentation to enrich the test image, filters out unreliable augmented images based on prediction entropy, and then updates learnable prompts by minimizing the entropy of the reliable predictions. DiffTPT (Feng et al., 2023) enhances TPT by augmenting an input image with generated images from a pre-trained diffusion model. C-TPT (Yoon et al., 2024) also enhances TPT by calibrating the prediction uncertainty. RLCF (Zhao et al., 2024b) introduces a CLIP score-based loss to avoid the pitfall of the entropy minimization. Because only a single unlabeled test image is available, existing methods leverage the outputs of VLMs on the test image for the adaptation. However, they solely rely on the *internal* knowledge encoded in the VLM parameters, which are constrained to the pre-training data (Agarwal et al., 2021; Menon et al., 2024; Parashar et al., 2024). In contrast, we leverage *external* knowledge retrieved from an external image database, which could have new or focused knowledge that is not present in the internal knowledge, thereby providing informative knowledge for unseen test images.

## 2.3 Retrieval-Augmented Strategy for Vision-Language Models

Since not all knowledge can be encoded within VLM parameters, retrieval-augmented strategies have been adopted as a solution, which can leverage external knowledge. These strategies can be broadly categorized into training-based and training-free methods. Because training-based methods exceed the scope of this work due to the necessity for extensive training and/or additional parameters before test time, we focus on training-free methods here. Training-free methods, including SuS-X (Udandarao et al., 2023), Neural Priming (Wallingford et al., 2023), and Ret-Adapter (Ming & Li, 2024)), retrieve training datasets and use the retrieved datasets to adapt VLMs without backpropagation based on few-shot adaptation methods. However, since they conduct the retrieval before test time, their retrieved images are static; thus, they cannot adaptively cope with unseen test images deviated by distribution shifts. On the other hand, we retrieve external images *adaptive* to test images on the fly and can flexibly handle distribution shifts. See Appendices A.1 and A.2 for the details of training-based and training-free methods, respectively.

## 2.4 Background on the CLIP Model

The contrastive language-image pre-trained (CLIP) model (Radford et al., 2021) consists of an image encoder $f(\cdot)$ and a text encoder $g(\cdot)$, each of which maps an image $x \in \mathbb{R}^{3 \times h \times w}$ and a text prompt $t$ into a $d$-dimensional shared embedding space, respectively. By calculating cosine similarity between the mapped embeddings $\mathbf{e}^I \in \mathbb{R}^d$ for the image and $\mathbf{e}^T \in \mathbb{R}^d$ for the text prompt, CLIP produces the image-text alignment score $s^{\text{align}} = \cos(\mathbf{e}^I, \mathbf{e}^T)$, which measures semantic similarity between the image $x$ and the text prompt $t$. For text prompts, text descriptions that represent the fine-grained visual features of objects can be utilized. For each class $c$, multiple $L$ text descriptions $\{t_{c,l}\}_{l=1}^{L}$ are generated and mapped to the embedding space. The embeddings of these descriptions for each class $c$ are averaged, and the averaged embedding $\bar{\mathbf{e}}_c^T$ is used as a prototype for the class $c$. Using the prototypes, the prediction probability for an image $x$ being a class $c$ can be calculated as

$$p(c \,|\, x) = \frac{\exp(\cos(\mathbf{e}^I, \bar{\mathbf{e}}_c^T)/\tau)}{\sum_{i=1}^{C} \exp(\cos(\mathbf{e}^I, \bar{\mathbf{e}}_i^T)/\tau)}, \tag{1}$$

where $C$ is the number of classes and $\tau$ is a temperature parameter.

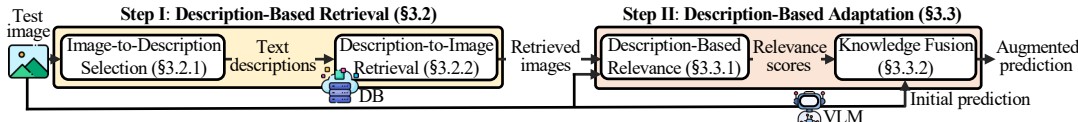

Figure 2: Overall procedure of RA-TTA.

# 3 RA-TTA: RETRIEVAL-AUGMENTED TEST-TIME ADAPTATION FOR VLMS

## 3.1 PROBLEM STATEMENT AND OVERVIEW

**Problem Statement.** RA-TTA aims to adapt a VLM (e.g., CLIP) to classify a test image $x^{\text{test}}$ as its label $y$ by leveraging a set of $N$ external image embeddings $\mathcal{R} = \{\mathbf{e}_k^{I^e} \mid k \in \mathcal{L}\}$ retrieved from an image vector database $\mathcal{D} = \{\mathbf{e}_j^{I^e}\}_{j=1}^D$, where $\mathcal{L}$ is the index set of the $N$ retrieved embeddings, $D$ is the database size, and $\mathbf{e}_j^{I^e} \in \mathbb{R}^d$ is the image embedding computed by an image encoder $f(\cdot)$ for an external image $x_j^{\text{e}}$.[1] In line with TPT (Shu et al., 2022), the training data is unavailable, the training pipeline cannot be modified, and the adaptation is conducted online on a single image.

**Overview.** Figure 2 illustrates the retrieval-and-adaptation procedure of RA-TTA. In **Step I**, for a test image, *description-based retrieval* selects relevant text descriptions and retrieves external images aligned with the selected text descriptions (§ 3.2). In **Step II**, *description-based adaptation* calculates relevance scores for the retrieved images with respect to the test image, and the initial prediction is refined by the relevance scores to produce the augmented prediction (§ 3.3). Its pseudo-docode is presented in Appendix B.

## 3.2 STEP I: DESCRIPTION-BASED RETRIEVAL

Figure 3 details the procedure of the description-based retrieval step. The retrieved images should include the visible features of the target object in a test image $x^{\text{test}}$, which are pivotal for recognizing the corresponding label $y$. To achieve this goal, multiple fine-grained visual features of each target class are extracted offline by LLMs in the form of text descriptions, which can be represented as $\mathcal{T} = \cup_{c=1}^C \{t_{c,l}\}_{l=1}^L$. Based on the descriptions in $\mathcal{T}$, we present a *description-based retrieval* approach to retrieve images that include the pivotal features of $x^{\text{test}}$ from $\mathcal{D}$. The bi-modality of VLMs is fully exploited first through image-to-text search (*Left* of Figure 3, § 3.2.1) and then through text-to-image search (*Right* of Figure 3, § 3.2.2).

### 3.2.1 IMAGE-TO-DESCRIPTION SELECTION (LEFT OF FIGURE 3)

**Robust Image-Text Alignment.** The image-text alignment scores between a test image $x^{\text{test}}$ and the text descriptions in $\mathcal{T}$ are calculated for selecting relevant text descriptions. Misleading descriptions may be inaccurately aligned with the test image, mainly due to non-target objects, likely leading to the retrieval of irrelevant images. To select only relevant descriptions without misleading ones, we enable a VLM to analyze a test image from multiple perspectives, thereby understanding it comprehensively. Specifically, the standard augmentation $A(\cdot)$, including random resized cropping and random flipping, is applied to a test image $x^{\text{test}}$, obtaining the augmented images $\mathcal{A} = \{x_m\}_{m=0}^M$, where $M$ is the number of augmented images by $A(\cdot)$ and the original test image $x^{\text{test}}$ is denoted as $x_0$. Subsequently, the alignment scores of each description $t_{c,l}$ are calculated with the images in $\mathcal{A}$. That is, $\mathcal{S}_{t_{c,l}} = \{s_{m,t_{c,l}}^{\text{align}} \mid x_m \in \mathcal{A}\}$, where $s_{m,t_{c,l}}^{\text{align}} = \cos(\mathbf{e}_m^I, \mathbf{e}_{c,l}^T)$, $\mathbf{e}_m^I = f(x_m)$, and $\mathbf{e}_{c,l}^T = g(t_{c,l})$. Although misleading descriptions may result in inaccurately elevated alignment scores for a few certain augmentations, the majority of augmentations overlook these misleading descriptions; the scores of the misleading descriptions are generally low, with only a few exceptions. Thus, to avoid selecting the misleading descriptions (i.e., features), the $p$-percentile of the scores in $\mathcal{S}_{t_{c,l}}$ is used as the *representative* image-text alignment score for the description $t_{c,l}$, which is represented by

$$s_{t_{c,l}}^{\text{align}} = \text{percentile}_p(\mathcal{S}_{t_{c,l}}) = \text{percentile}_p(\{s_{m,t_{c,l}}^{\text{align}}\}_{m=0}^M), \tag{2}$$

where $\text{percentile}_p(\cdot)$ returns the $100p$-th ($p \in [0,1]$) percentile of a set. We choose $p = 0.75$ or $0.90$, with which the function returns the top 75% (i.e., *third quartile (Q3)*) or 90% value of $\mathcal{S}_{t_{c,l}}$.

---

[1]For simplicity, we refer to $\mathbf{e}^I$ and $\mathbf{e}^T$ as an image and a text description, respectively, omitting the term "embedding" hereafter.

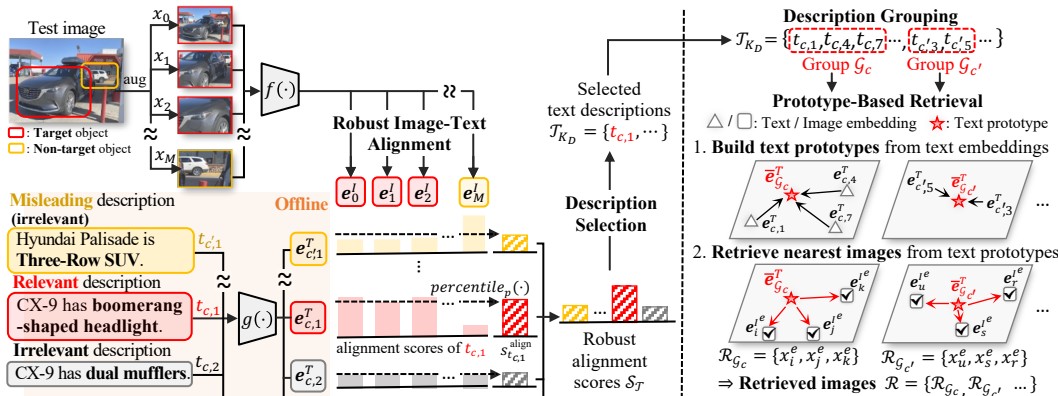

Figure 3: **Description-based retrieval**. *Left*: Image-to-description selection. A test image is augmented to produce multiple views, and the image-text alignment scores between the augmented images and the text descriptions are calculated. The top-$K$ descriptions are selected based on the third quartile (Q3) value of the alignment scores of each description to avoid misleading and irrelevant descriptions. *Right*: Description-to-image retrieval. The selected descriptions are grouped by their corresponding class, and those in each group are embedded and averaged to build a text prototype. Finally, the external images closest to the prototypes are retrieved.

Consequently, the robust alignment scores of the $x^{\text{test}}$ with the text descriptions in $\mathcal{T}$ become

$$\mathcal{S}_{\mathcal{T}} = \{s_{t_{c,l}}^{\text{align}} \mid t_{c,l} \in \mathcal{T}\}. \tag{3}$$

**Description Selection.** Since the high image-text alignment score of an image-text pair represents the high correspondence between them, the top $K_D$ descriptions in $\mathcal{T}_{K_D}$ are selected as the pivotal features of $x^{\text{test}}$ using the alignment scores in $\mathcal{S}_{\mathcal{T}}$, which can be represented by

$$\mathcal{T}_{K_D} = \{t_{c,l} \mid s_{t_{c,l}}^{\text{align}} \in \text{Top-K}(\mathcal{S}_{\mathcal{T}}; K_D)\}, \tag{4}$$

where Top-K$( \cdot ; K)$ is a function that selects the $K$ highest values from a set.

### 3.2.2 DESCRIPTION-TO-IMAGE RETRIEVAL (RIGHT OF FIGURE 3)

**Description Grouping.** Before conducting the retrieval, we arrange the selected descriptions because those of different classes could be mixed, where the corresponding classes of the descriptions in $\mathcal{T}_{K_D}$ are denoted as $\mathcal{C} = \{c \mid t_{c,l} \in \mathcal{T}_{K_D}\}$. The text descriptions in $\mathcal{T}_{K_D}$ are grouped by their corresponding class as $\mathcal{G} = \{\mathcal{G}_c\}_{c \in \mathcal{C}}$, where $\mathcal{G}_c$ includes those of a class $c$ and represents the semantics of the class $c$ with the features indicated by the belonging descriptions.

**Prototype-Based Retrieval.** Based on the descriptions in $\mathcal{G}_c$, we retrieve the external images that can be interpreted as representing the semantics of $\mathcal{G}_c$ and attach $c$ as their pseudo label. First, the embeddings of the descriptions in $\mathcal{G}_c$ are simply averaged to build a text prototype $\bar{\mathbf{e}}_{\mathcal{G}_c}^T$ that serves as the representative embedding of the semantics in $\mathcal{G}_c$. Then, the alignment scores between $\bar{\mathbf{e}}_{\mathcal{G}_c}^T$ and the embeddings of external images in $\mathcal{D} = \{\mathbf{e}_j^{I^e}\}_{j=1}^D$ are calculated, and the top $K_S$ external images are retrieved using the alignment scores, which can be represented as

$$\mathcal{R}_{\mathcal{G}_c} = \{x_j^e \mid s_j^{\text{align}} \in \text{Top-K}(\mathcal{S}_{\mathcal{D}}; K_S)\}, \tag{5}$$

where $\mathcal{S}_{\mathcal{D}} = \{s_j^{\text{align}}\}_{j=1}^D$ and $s_j^{\text{align}} = \cos(\mathbf{e}_j^{I^e}, \bar{\mathbf{e}}_{\mathcal{G}_c}^T)$.[2] The same procedure is applied separately to each group $\mathcal{G}_c$. Since the same number ($K_S$) of images are retrieved for each group, the total number of retrieved images is $N = K_S \times |\mathcal{C}|$. Finally, $\mathcal{R} = \{\mathcal{R}_{\mathcal{G}_c}\}_{c \in \mathcal{C}}$ is fed to the next step.

### 3.3 STEP II: DESCRIPTION-BASED ADAPTATION

By leveraging the set of retrieved external images, $\mathcal{R} = \{\mathcal{R}_{\mathcal{G}_c}\}_{c \in \mathcal{C}}$, the VLM's initial prediction for a test image $x^{\text{test}}$ is adjusted by an additional prediction based on $\mathcal{R}$, where the initial prediction $p$ is conducted as in Eq. (1). Since each set of external images, $\mathcal{R}_{\mathcal{G}_c}$, is associated with its *pseudo label* $c$, we first calculate the relevance score from $x^{\text{test}}$ to the set of external images, $\mathcal{R}_{\mathcal{G}_c}$, which is then used to determine the probability of $x^{\text{test}}$ belonging to the corresponding class $c$. Interestingly, to calculate this relevance score, the selected text descriptions, $\mathcal{T}_{K_D}$ in Eq. (4), are exploited again

---

[2]K-nearest neighbor search can be performed very efficiently with conventional vector databases.

(§ 3.3.1), where our *description-based adaption* kicks in. Subsequently, both initial and additional predictions are fused to produce the augmented prediction (§ 3.3.2). In short, this adaptation step can be viewed as an ensemble procedure in which each retrieved external image casts a vote for the label of $x^{\text{test}}$ based on its pseudo label.

### 3.3.1 DESCRIPTION-BASED RELEVANCE

**Semantic Gap.** Since we use the pivotal features (i.e., text descriptions) of a test image to retrieve external images, it is logical to employ these descriptions to assess the relevance between a test image and an external image. We intend to measure the disparity between two images concerning a specific semantic aspect. We specifically introduce the *semantic gap* between two images, which pertains to the specific semantic aspect, in Definition 3.1. As illustrated in Figure 4, it is the difference between the distances from each image embedding to the embedding representing the semantic aspect of interest. For instance, when examining the front grille of vehicles, the semantic gap would signify the disparity between two vehicle images in terms of the clarity with which they depict the front grille.

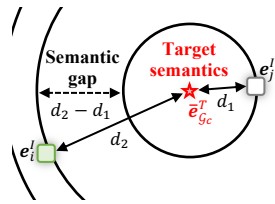

Figure 4: Semantic gap.

**Definition 3.1** (SEMANTIC GAP). Given the descriptions in $\mathcal{G}_c$, the *semantic gap* between two images $x_i$ and $x_j$ is the absolute difference between their cosine distances with the text prototype $\bar{\mathbf{e}}^T_{\mathcal{G}_c}$ of the descriptions in $\mathcal{G}_c$ in the embedding space of a VLM,

$$\text{gap}(x_i, x_j, \mathcal{G}_c) = |(1 - \cos(\mathbf{e}^I_i, \bar{\mathbf{e}}^T_{\mathcal{G}_c})) - (1 - \cos(\mathbf{e}^I_j, \bar{\mathbf{e}}^T_{\mathcal{G}_c}))|, \tag{6}$$

where $\mathbf{e}^I_i = f(x_i)$, $\mathbf{e}^I_j = f(x_j)$, and $\bar{\mathbf{e}}^T_{\mathcal{G}_c}$ is the text prototype of the descriptions in $\mathcal{G}_c$ computed by averaging their embeddings.

**Semantic Relevance Computation.** Through the concept of the semantic gap, we now derive the relevance score of a test image to the set of external images for a specific class. To enhance its reliability and robustness, we devise two simple yet effective techniques. First, we reuse the set $\mathcal{A}$ of augmented images for the test image $x^{\text{test}}$ instead of considering it only. As a result, the relevance score is aggregated from the pair-wise semantic gaps between $(M + 1)$ augmented images and $K_S$ external images. Second, the significance of each image within each set, i.e., $\mathcal{A}$ or $\mathcal{R}_{\mathcal{G}_c}$, is factored into the relevance score. If an image is located closer to the text prototype of the corresponding set in the embedding space of a VLM, the image should be considered more importantly in aggregating the pairwise semantic gaps. Mathematically, a matrix $\mathbf{C}_{\mathcal{G}_c} \in \mathbb{R}^{(M+1) \times K_S}$ has the pair-wise semantic gaps between $x_i \in \mathcal{A}$ and $x^e_j \in \mathcal{R}_{\mathcal{G}_c}$, which can be represented as

$$\mathbf{C}_{\mathcal{G}_c} = \left[ \text{gap}(x_i, x^e_j, \mathcal{G}_c) \mid_{x_i \in \mathcal{A}, x^e_j \in \mathcal{R}_{\mathcal{G}_c}} \right] \in \mathbb{R}^{(M+1) \times K_S}. \tag{7}$$

For the significance of each image, we additionally define two weight vectors $\mathcal{U} \in \mathbb{R}^{(M+1)}$ and $\mathcal{V} \in \mathbb{R}^{K_S}$: $\mathcal{U}$ contains normalized weights from the cosine similarity between the embedding of $x_i \in \mathcal{A}$ and the prototype $\bar{\mathbf{e}}^T_{\mathcal{A}}$ of $\mathcal{A}$, where $\bar{\mathbf{e}}^T_{\mathcal{A}}$ is calculated by averaging the embeddings of the descriptions in $\mathcal{T}_{K_D}$ (see Eq. (4)); $\mathcal{V}$ contains normalized weights from the cosine similarity between the embedding of $x_j \in \mathcal{R}_{\mathcal{G}_c}$ and $\bar{\mathbf{e}}^T_{\mathcal{G}_c}$ (see Eq. (6)). Considering these weight vectors, the pair-wise semantic gaps in $\mathbf{C}_{\mathcal{G}_c}$ are aggregated by the optimal transport (OT) framework (Villani et al., 2009), which can measure the distance between the two sets of weighted data points. Then, the relevance score to a class $c$ is formulated by

$$s^{\text{rel}}_{\mathcal{G}_c} = \frac{1}{\text{OT}_{\text{dist}}(\mathbf{C}_{\mathcal{G}_c}, \mathcal{U}, \mathcal{V}) + 1}, \tag{8}$$

where $\text{OT}_{\text{dist}}(\cdot, \cdot, \cdot)$ returns the aggregated distance based on the OT framework. The same procedure repeats for each group $\mathcal{R}_{\mathcal{G}_c}$ (i.e., for each class), resulting in $\mathcal{RS} = \{s^{\text{rel}}_{\mathcal{G}_c} \mid c \in \mathcal{C}\}$. We further provide the details of OT in Appendix C.

### 3.3.2 KNOWLEDGE FUSION FOR ADAPTATION

We derive the retrieval-based prediction $\hat{p}$ for $x^{\text{test}}$ using the relevance scores in $\mathcal{RS}$, and the prediction probability for the image $x^{\text{test}}$ of being a class $c$ can be represented as

$$\hat{p}(c \mid x^{\text{test}}) = \begin{cases} \dfrac{\exp(s^{\text{rel}}_{\mathcal{G}_c})/\tau)}{\sum_{c \in \mathcal{C}} \exp(s^{\text{rel}}_{\mathcal{G}_c})/\tau)} & \text{if } c \in \mathcal{C} \\ 0 & \text{otherwise} \end{cases}. \tag{9}$$

Finally, we fuse the retrieval-based prediction $\hat{p}$ and the initial prediction $p$ using the entropy of each prediction, leading to the augmented prediction $p^{\text{aug}}$ for the image $x^{\text{test}}$,

$$p^{\text{aug}}(c \,|\, x^{\text{test}}) = \alpha \times p(c \,|\, x^{\text{test}}) + (1 - \alpha) \times \hat{p}(c \,|\, x^{\text{test}}). \tag{10}$$

Here, $\alpha = \frac{\exp(1/(1+\text{H}))}{\exp(1/(1+\text{H})) + \exp(1/(1+\hat{\text{H}}))}$, where H (or $\hat{\text{H}}$) is the entropy of $P$ (or $\hat{P}$).

# 4 EVALUATION

## 4.1 EXPERIMENT SETTING

**Datasets.** To evaluate the zero-shot transferability, we use standard transfer learning and natural distribution shift benchmarks, including 17 datasets that span a wide range of image classification tasks: ImageNet (Deng et al., 2009), Flowers102 (Nilsback & Zisserman, 2008), DTD (Cimpoi et al., 2014), Oxford pets (Parkhi et al., 2012), Stanford cars (Krause et al., 2013), UCF101 (Soomro et al., 2012), Caltech101 (Fei-Fei et al., 2004), Food101 (Bossard et al., 2014), SUN397 (Xiao et al., 2010), FGVC aircraft (Maji et al., 2013), RESISC45 (Cheng et al., 2017), Caltech256 (Griffin et al., 2007), and CUB200 (Wah et al., 2011) as transfer learning benchmarks, and natural distribution shift benchmarks, including ImageNet adversarial (Hendrycks et al., 2021b), ImageNet V2 (Recht et al., 2019), ImageNet rendition (Hendrycks et al., 2021a), and ImageNet sketch (Wang et al., 2019). The details of each dataset can be found in Appendix D.1.

**Compared Methods.** We compare RA-TTA against four kinds of VLM adaptation methods that do not require training procedures before test time: (1) zero-shot CLIP baselines, (2) tuning-based methods, (3) text description-based methods, and (4) retrieval-based methods. For zero-shot CLIP baselines, we include two zero-shot baselines of CLIP using a default prompt, "a photo of a {class name}," and the ensemble of 80 hand-crafted prompts (Radford et al., 2021). For tuning-based methods, we use TPT (Shu et al., 2022), C-TPT (Yoon et al., 2024), and RLCF (Zhao et al., 2024b), which update parameters through back-propagation from VLM outputs for a test image. We use VisDesc (Menon & Vondrick, 2022), WaffleCLIP (Roth et al., 2023), and CuPL (Pratt et al., 2023) as text description-based methods that leverage the contextual richness of language. For retrieval-based methods, we include SuS-X-LC (Udandarao et al., 2023) and Neural Priming (Wallingford et al., 2023), which retrieve external images from an image database *offline*.

**Web-Scale External Image Database Construction.** We construct the database for retrieval-based methods, including SuS-X-LC, Neural Priming, and our proposed RA-TTA, with the following objectives: ensuring relevance to downstream tasks and preserving the diversity and noisy nature of a web-scale database. Thus, we explore LAION2B (Schuhmann et al., 2022) dataset that consists of 2B web-scale image-caption pairs for the construction. Following keyword-based fast string matching on LAION2B (Wallingford et al., 2023; Parashar et al., 2024), we downloaded images whose text captions contain the target classes and used the downloaded images as external images. This approach offers several advantages: it ensures the relevance and preserves the nature of a web-scale database (Parashar et al., 2024), not to mention reproducibility. While using all images in LAION2B may seem appealing, it requires 100 TB of storage. We conjecture that our database achieves a practical balance between leveraging web-scale images and the complexity of evaluation. The details for the image database construction can be found in Appendix D.2

**Implementation Details.** We implement RA-TTA using the CLIP-B/16 (Radford et al., 2021) model as a VLM. For generating text descriptions, we use GPT-3.5 Turbo (Ouyang et al., 2022) and CuPL (Pratt et al., 2023)-style hand-written templates. We set the augmentation size $M$ to 100. We configure $K$ for Top-$K$ operations at $K_D = 20$ and $K_S = 20$. We use a temperature parameter $\tau$ of 0.01, which is the default scale value of CLIP. We use the aforementioned configurations across all datasets because dataset-specific configurations are not preferred in TTA. We set $p = 0.75$ for transfer learning datasets except ImageNet and $p = 0.90$ for ImageNet-based datasets. Further implementation details on the compared methods can be found in Appendix D.3. All implementations are conducted using PyTorch 2.3.0 on an NVIDIA RTX 4090. The source code of RA-TTA is available at `https://github.com/kaist-dmlab/RA-TTA`.

## 4.2 MAIN RESULTS

Table 1 shows a comparative analysis of RA-TTA against the compared methods for 13 transfer learning benchmarks, evaluating how much the given methods can improve the zero-shot capability

Table 1: **Overall results for test-time adaptation (or transfer learning).** We report the top-1 accuracy (%) for 13 standard transfer learning datasets. The "Avg." column indicates the average accuracy of 13 datasets. The best results are in **bold**, and the second best results are underlined.

| | IN-1k | Flowers102 | DTD | Oxford pets | Stanford cars | UCF101 | Caltech101 | Food101 | SUN397 | FGVC aircraft | RESISC45 | Caltech256 | CUB200 | Avg. (13) |
|---|---|---|---|---|---|---|---|---|---|---|---|---|---|---|
| CLIP | 66.76 | 67.19 | 44.50 | 88.14 | 65.27 | 64.92 | 92.78 | 85.40 | 62.55 | 24.60 | 55.70 | 82.80 | 58.08 | 66.05 |
| Ensemble | 68.37 | 65.85 | 45.21 | 88.20 | 66.34 | 67.41 | 93.77 | 85.41 | 65.79 | 24.39 | 58.35 | 85.81 | 58.61 | 67.19 |
| TPT | 69.08 | 69.18 | 47.04 | 87.44 | 66.55 | 68.04 | 93.79 | 86.34 | 65.32 | 23.31 | 56.84 | 85.37 | 60.11 | 67.57 |
| C-TPT | 68.32 | 69.43 | 45.27 | 88.25 | 65.48 | 65.50 | 93.39 | 84.95 | 64.55 | 24.39 | 56.02 | 85.25 | 58.84 | 66.90 |
| RLCF | 68.61 | 67.72 | 46.40 | 86.73 | 66.51 | 66.98 | 93.83 | 86.09 | 64.92 | 23.43 | 56.89 | 85.18 | 57.91 | 67.02 |
| VisDesc | 69.09 | 71.86 | 50.41 | 88.55 | 65.48 | 69.52 | 94.81 | 86.43 | 68.25 | 25.59 | 57.81 | 88.17 | 60.13 | 68.93 |
| WaffleCLIP | 69.05 | 72.59 | 48.33 | 89.79 | 64.60 | 69.13 | 94.61 | 86.85 | 67.17 | 25.25 | 63.31 | 88.10 | 59.83 | 69.12 |
| CuPL | 69.78 | 75.92 | 58.22 | 91.47 | 66.92 | 67.80 | 94.24 | 86.39 | 67.38 | 28.98 | 65.17 | 88.07 | 60.18 | 70.81 |
| SuS-X-LC | 69.45 | 76.23 | 59.23 | 91.83 | 67.55 | 67.12 | 93.78 | 86.13 | 67.78 | 29.41 | 65.22 | 88.75 | 59.12 | 70.89 |
| Neural Priming | 69.38 | 73.22 | 55.98 | 89.76 | 66.13 | 68.02 | 94.71 | 87.01 | 67.86 | 27.32 | 63.11 | 88.50 | 57.14 | 69.86 |
| **RA-TTA (Ours)** | **70.58** | **78.65** | **60.98** | **92.78** | **70.11** | **73.28** | **94.84** | **87.10** | **70.38** | **32.34** | **66.95** | **89.50** | **62.73** | **73.09** |

of VLMs. RA-TTA outperforms all existing methods, with an average improvement of 3.01–9.63% over the baselines, demonstrating its effectiveness in enhancing the knowledge of VLMs at test time. The tuning-based methods (Shu et al., 2022; Yoon et al., 2024; Zhao et al., 2024b) cannot leverage knowledge from an external database and the contextual information from textual modality, thus failing to adapt. On the other hand, RA-TTA achieves the best performance by leveraging the external knowledge and the textual information, even outperforming text description-based methods (Menon & Vondrick, 2022; Pratt et al., 2023; Roth et al., 2023) that adopt fine-grained text descriptions but cannot use external knowledge unlike RA-TTA.

RA-TTA is particularly effective for a specialized domain dataset, like RESISC45 for satellite images, and fine-grained datasets, such as Flowers102, Stanford Cars, FGVC Aircraft, and CUB200 because RA-TTA retrieves a *customized* set of external images for each test image by identifying its pivotal features through fine-grained descriptions. In contrast, previous retrieval-based methods, including SuS-X-LC (Udandarao et al., 2023) and Neural Priming (Wallingford et al., 2023) retrieve a *fixed* set of external images for a given class, which limits the granularity of extractable knowledge and ultimately results in inferior performance compared to RA-TTA.

Moreover, Table 2 illustrates the performance of RA-TTA for ImageNet variants (Hendrycks et al., 2021b; Recht et al., 2019; Hendrycks et al., 2021a; Wang et al., 2019) which are typically used to evaluate the VLMs' robustness to natural distribution shifts, such as artistic rendition and black and white sketches (Radford et al., 2021; Shu et al., 2022). RA-TTA outperforms the compared methods in terms of the average accuracy, demonstrating that RA-TTA can cope with natural distribution shifts effectively. Notably, the description-based methods (e.g., CuPL (Pratt et al., 2023)) that employ only fine-grained descriptions *without* help from external knowledge show minor

Table 2: **Performance for natural distribution shifts.** We report the top-1 accuracy (%) for four ImageNet variants. The "Avg." column indicates the average accuracy of 4 datasets.

| | IN-A | IN-V2 | IN-R | IN-K | Avg. (4) |
|---|---|---|---|---|---|
| CLIP | 47.51 | 60.80 | 73.98 | 46.19 | 57.12 |
| Ensemble | 50.04 | 61.89 | 77.58 | 48.29 | 59.45 |
| TPT | 54.39 | 63.48 | 77.27 | 47.95 | 60.77 |
| C-TPT | 50.28 | 62.47 | 75.68 | 47.42 | 58.96 |
| RLCF | 56.52 | 63.37 | 77.04 | 48.09 | 61.26 |
| VisDesc | 50.17 | 62.76 | 75.25 | 48.25 | 59.11 |
| WaffleCLIP | 50.51 | 62.68 | 75.81 | 48.73 | 59.43 |
| CuPL | 50.23 | 63.00 | 78.16 | 49.60 | 60.25 |
| SuS-X-LC | 49.91 | 63.22 | 77.82 | 49.18 | 60.03 |
| Neural Priming | 49.68 | 62.79 | 76.70 | 49.03 | 59.55 |
| **RA-TTA (Ours)** | **59.21** | **64.16** | **79.68** | **50.83** | **63.47** |

improvements compared to Ensemble (Radford et al., 2021). However, RA-TTA improves the accuracy by 15.49% over Ensemble, indicating that the effectiveness of RA-TTA arises not only from the use of fine-grained descriptions but also from the sophisticated integration of external knowledge. The standard deviations for Tables 1 and 2 are presented in Appendix E.1 owing to the lack of space.

### 4.3 ABLATION STUDY

In Table 3, we conduct ablation studies to understand the impact of the description-based retrieval, the description-based adaptation, and the image weighting scheme. (1) If the description-based

retrieval is disabled, the retrieval procedure in Section 3.2 is simply replaced by image-to-image similarity search. (2) If the description-based adaptation is disabled, the semantic gap in Definition 3.1 is simply replaced by image-to-image cosine similarity. (3) If the image weighting scheme is disabled, the weight vectors $\mathcal{U}$ and $\mathcal{V}$ in Eq. (8) are not used for aggregating the pair-wise distances. We prepare three different variants, denoted by Var. 1, Var. 2, and Var. 3 in Table 3, by disabling three, two, and one component(s), respectively. While the description-based retrieval is effective by itself (compare between Var. 1 and Var. 2), its effect is clearly boosted when combined with the description-based adaptation (compare between Var. 2 and Var. 3). In addition, the image weighting scheme adds a slight improvement (compare between Var. 3 and RA-TTA). Overall, all three components are shown to be effective, supporting the superior performance of RA-TTA. The results for other datasets can be found in Appendix E.2. Furthermore, refer to Appendices E.3 and E.4 about the results of changing the LLM for generating descriptions and varying the database size.

Table 3: **Ablation studies.** We report the top-1 accuracy (%) on the FGVC aircraft dataset, where the benefit of RA-TTA is significant. Description-based retrieval, description-based adaptation, and image weighting are disabled separately.

|  | Retrieval | Adaptation | Weighting | Accuracy |
|---|---|---|---|---|
| Var. 1 | ✗ | ✗ | ✗ | 29.39 |
| Var. 2 | ✓ | ✗ | ✗ | 30.91 |
| Var. 3 | ✓ | ✓ | ✗ | 31.96 |
| **RA-TTA** | ✓ | ✓ | ✓ | **32.34** |

## 4.4 HYPERPARAMETER SENSITIVITY ANALYSIS

We conduct sensitivity analyses on four crucial hyperparameters which influence the performance of RA-TTA. The results in Figure 5 report the averaged accuracy on 13 standard transfer learning benchmarks. Refer to Appendix E.5 for more results.

- **Effect of Augmentation Size.** For mitigating the interfering effect of misleading descriptions, RA-TTA conducts augmentations to analyze a test image from multiple perspectives. We analyze the impact of the augmentation size $M$ on the accuracy. As shown in Figure 5(a), increasing the augmentation size improves the accuracy, with a plateau observed around 25. This result suggests that while more augmentations are generally beneficial, the benefits diminish beyond a certain point (around 25), where the accuracy stabilizes.

- **Effect of $K_D$ for Description Selection.** RA-TTA selects the top $K_D$ descriptions as the pivotal features of a test image. Figure 5(b) demonstrates the accuracy concerning the $K_D$. RA-TTA performs best at $K_D = 20$, after which the accuracy declines sharply. This result can be understood as increased $K_D$ may lead to selecting irrelevant or misleading descriptions, negatively impacting the following retrieval and adaptation.

- **Effect of $K_S$ for Image Retrieval.** RA-TTA retrieves the top $K_S$ images using the selected text descriptions. Figure 5(c) illustrates the effect of $K_S$. As for $K_D$, we can infer that the accuracy drops because outlier images can be included as $K_S$ increases. However, the accuracy remains stable across different values of $K_S$, with no significant drop even as $K_S$ increases. This result suggests that when aggregating the semantic gap, representing the significance weights of retrieved images makes RA-TTA robust to the retrieved outlier images.

- **Effect of Alignment Score Percentile.** Figure 5(d) shows the accuracy based on the percentile value $p$ for rejecting misleading descriptions. The best accuracy is achieved around 0.75, the third quartile value, which is optimal for selecting relevant descriptions while rejecting misleading descriptions. When the value is set too high (e.g., 1.0), the accuracy degrades, possibly due to the inclusion of misleading descriptions.

## 4.5 EFFICIENCY ANALYSIS

In the context of TTA, the efficiency is indeed a critical factor to consider. We provide an analysis of computational complexity, including inference time and GPU

Table 4: GPU inference time per sample (s/sample)

|  | FGVC aircraft | Stanford cars | RESISC45 | Avg. (3) |
|---|---|---|---|---|
| TPT | **0.103** | 0.155 | **0.95** | 0.118 |
| **RA-TTA (Ours)** | 0.113 | **0.117** | 0.121 | **0.117** |

peak memory. Based on Table 4, which shows the GPU inference time per sample (i.e., s/sample), the inference time of RA-TTA is comparable to that of TPT, the representative TTA method. This is because FAISS (Johnson et al., 2021), an advanced and efficient search engine typically used for RAG-based approaches, enables fast nearest-neighbor search. The analysis of GPU peak memory is provided in Appendix E.6 owing to the lack of space.

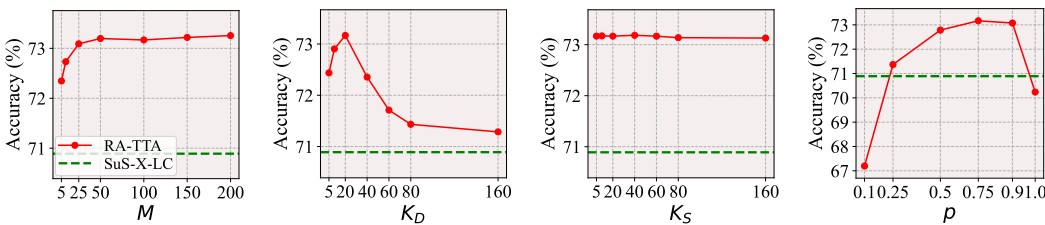

(a) Augmentation size.  (b) # of selected descriptions.  (c) # of retrieved images.  (d) Score percentile.

Figure 5: Effects of the four hyperparameters on the accuracy. The green dashed line denotes the accuracy of SuS-X-LC as a strong baseline.

## 4.6 QUALITATIVE ANALYSIS

In Figure 6, we present illustrative examples of the description-based retrieval of RA-TTA on the Stanford cars (Krause et al., 2013) and RESISC45 Cheng et al. (2017) datasets. We observe in Figure 6 that description-based retrieval allows us to retrieve external images with the pivotal features of test images because the retrieval procedure focuses on a specific piece of information, i.e., a pivotal feature, without being distracted by other irrelevant information. Additional visualizations are provided in Appendix E.7.

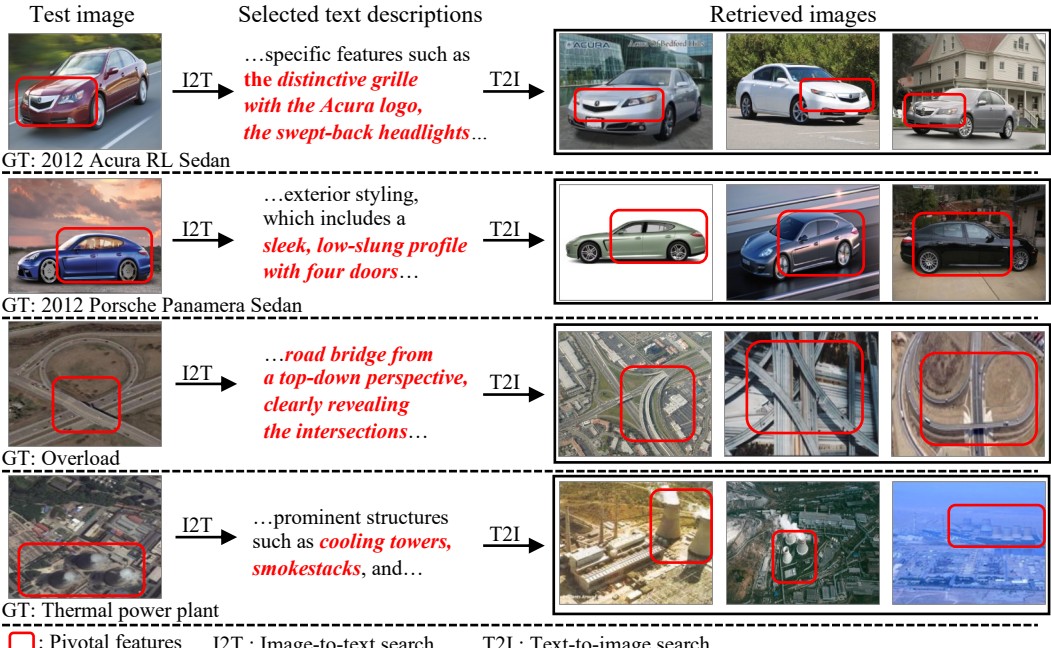

Figure 6: Examples of the images retrieved by our description-based retrieval on the Stanford cars and RESISC45 datasets. The results of the image-to-description selection and the description-to-image retrieval are shown individually. A pivotal feature is indicated by a red box in the images.

## 5 CONCLUSION

In this paper, we propose RA-TTA (Retrieval-Augmented Test-Time Adaptation), designed to leverage the external knowledge from a web-scale image database on-the-fly. By fully exploiting the text descriptions, which represent fine-grained visual details, the *description-based retrieval* ensures to retrieve external images with the pivotal features of a test image, and the *description-based adaptation* allows a VLM to adapt to the test image using the inherent semantics of the retrieved images. Our extensive experimental results demonstrate superior zero-shot transferability compared to the state-of-the-art methods across 17 datasets. Overall, we believe that our work sheds light on the effectiveness of external knowledge for the zero-shot transfer of VLMs.

## ETHICS STATEMENT

This work adheres to the ethical guidelines and principles of ICLR, ensuring that no harm was done to individuals, groups, or environments during the study. A web-scale external image database may contain problematic images if it is constructed carelessly. Even though we use a previous version of LAION2B as the data source, we do *not* include the known problematic images, because normal, unproblematic images are selected by querying the class names of our downstream tasks. Since a cleaned version of LAION2B has been released just a month ago, to further strengthen our ethical standard, we will reconstruct our external image database using the cleaned version.

## REPRODUCIBILITY STATEMENT

In accordance with ICLR's reproducibility guidelines, we provide a GitHub repository containing the code used for our experiments: `https://github.com/kaist-dmlab/RA-TTA`.

## ACKNOWLEDGEMENT

This work was supported by Institute of Information & Communications Technology Planning & Evaluation (IITP) grant funded by the Korea government (MSIT) (No. RS-2020-II200862, DB4DL: High-Usability and Performance In-Memory Distributed DBMS for Deep Learning, 50% and No. RS-2022-II220157, Robust, Fair, Extensible Data-Centric Continual Learning, 50%).

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

## A  EXTENDED SURVEY

### A.1  TRAINING-BASED RETRIEVAL-AUGMENTED METHODS FOR VLMS

Within retrieval-augmented strategies for VLMs, training-based methods have proposed (additional) pre-training schemes which make use of external knowledge as training datasets (Shen et al., 2022; Liu et al., 2023) and which introduce specifically designed modules for external knowledge (Xie et al., 2023; Iscen et al., 2023). K-LITE (Shen et al., 2022) augments text supervisions by incorporating external textual knowledge such as Wiktionary and uses these enhanced supervisions to train VLMs. REACT (Liu et al., 2023) fine-tunes VLMs by constructing datasets about downstream tasks through retrieval. RA-CLIP (Xie et al., 2023) and RECO (Iscen et al., 2023) introduce additional modules that allow VLMs to utilize retrieved images, where the parameters are trained using large-scale image-text datasets (e.g., WebLI (Chen et al., 2023), YFCC (Thomee et al., 2016), and Conceptual Captions 12M (Changpinyo et al., 2021)). However, these methods require extensive training before test time, which demands high computational resources, making them challenging to use for resource-constrained applications. On the other hand, RA-TTA requires neither training nor additional parameters, hence facilitating its straightforward implementation.

### A.2  TRAINING-FREE ADAPTATION METHODS FOR VLMS

Training-free adaptation methods for VLMs (Zhang et al., 2022; 2023; Zhu et al., 2023; Wang et al., 2024b) have aimed to adapt VLMs to downstream tasks without updating learnable parameters. They can conduct the adaptation efficiently because they do not require backpropagation. Tip-Adapter (Zhang et al., 2022) introduces a non-parametric cache model that stores the embeddings of few-shot training images and the corresponding labels. Next, it computes the similarities between the embeddings in the cache model and a test image, uses them as a weight to aggregate the labels in the cache model, and integrates the aggregated pseudo-labels with the logits of CLIP. In order to improve Tip-Adapter, CaFo (Zhang et al., 2023) adds a vision foundation model called DINO, synthetic images from DALL-E (Ramesh et al., 2021) to expand the few-shot training data, and text descriptions generated by GPT-3 to enrich the language context. APE (Zhu et al., 2023) improves Tip-Adapter by extracting useful features from the cache model, and CLIP-GDA (Wang et al., 2024b) applies the Gaussian Discriminant Analysis (GDA) to create a feature-based classifier. However, these methods only focus on adapting VLMs without backpropagation and do not account for using external knowledge. In contrast, the goal of RA-TTA is leveraging *external knowledge* to complement the absence in the pre-trained knowledge, which is orthogonal to the objective of the training-free methods.

## B  PSEUDOCODE OF RA-TTA

The overall procedure of RA-TTA is described in Algorithm 1, which is self-explanatory.

## C  OPTIMAL TRANSPORT FOR CALCULATING SEMANTIC RELEVANCE SCORE

Optimal Transport (OT) (Villani et al., 2009) is a framework for calculating the distance between two distributions, typically represented by sampled data points. Mathematically, each distribution is formulated as

$$U = \sum_{i=1}^{M} u_i \delta_{x_i} \quad \text{and} \quad V = \sum_{j=1}^{N} v_j \delta_{y_j}, \tag{11}$$

where $x_i$ is a sample from $U$, $u_i$ is a probability mass of $x_i$, i.e., $\sum_{i=1}^{M} u_i = 1$, and $\delta_{x_i}$ is a Dirac delta function placed at $x_i$. The same definition applies for $V$. Here, $M$ and $N$ denote the number of data points sampled from $U$ and $V$, respectively. When calculating the distance, OT requires a disparity matrix $C$ representing pairwise disparities between samples from the two distributions. Then, to find the optimal transportation plan $\Gamma^*$, the optimization problem of OT is formulated as

$$\min_{\Gamma \in \Pi(\mathbf{u}, \mathbf{v})} \langle \Gamma, \mathbf{C} \rangle - \lambda H(\Gamma), \quad \Pi(\mathbf{u}, \mathbf{v}) = \left\{ \Gamma \in \mathbb{R}_+^{M \times N} \mid \Gamma \mathbf{1}_N = \mathbf{u}, \ \Gamma^\top \mathbf{1}_M = \mathbf{v} \right\}, \tag{12}$$

where $\mathbf{C} \in \mathbb{R}^{M \times N}$ is a disparity matrix where each element $c_{i,j}$ represents the disparity between $x_i$ and $y_j$, $\mathbf{u} \in \mathbb{R}^M$ is a probability mass vector whose $i$-th element is $u_i$, $\mathbf{v} \in \mathbb{R}^N$ is a probability mass

---

**Algorithm 1** RA-TTA: Overall Procedure

---

**Require:** a test image $x^{\text{test}}$, text descriptions $\mathcal{T} = \cup_{c=1}^{C} \{t_{c,l}\}_{l=1}^{L}$, an external image vector database $\mathcal{D}$, an image encoder $f(\cdot)$, a text encoder $g(\cdot)$, augmentation size $M$, K for description selection $K_D$, K for image retrieval $K_S$, percentile value $p$
**Ensure:** final prediction $p^{\text{aug}}$
 1: /* STEP I: DESCRIPTION-BASED RETRIEVAL IN § 3.2 */
 2: /* IMAGE-TO-DESCRIPTION SELECTION IN § 3.2.1 */
 3: $\mathcal{S}_{\mathcal{T}} \leftarrow \text{RobustAlignment}(x^{\text{test}}, \mathcal{T}, M, p, f, g)$
 4: $\mathcal{T}_{K_D} \leftarrow \text{DescriptionSelction}(\mathcal{S}_{\mathcal{T}}, K_D, \mathcal{T})$
 5: /* DESCRIPTION-TO-IMAGE RETRIEVAL IN § 3.2.2 */
 6: $\mathcal{G} \leftarrow \text{DescriptionGrouping}(\mathcal{T}_{K_D})$
 7: $\mathcal{R} \leftarrow \text{PrototypeBasedRetrieval}(\mathcal{G}, \mathcal{D}, K_S, g)$
 8: /* STEP II: DESCRIPTION-BASED ADAPTATION IN § 3.3 */
 9: /* DESCRIPTION-BASED RELEVANCE SCORE COMPUTATION IN § 3.3.1 */
10: $\mathbf{C} \leftarrow \text{SemanticGapComputation}(x^{\text{test}}, \mathcal{G}, \mathcal{R}, f, g)$
11: $\mathcal{RS} \leftarrow \text{RelevanceScoreComputation}(\mathbf{C}, \mathcal{T}_{K_D}, \mathcal{G})$
12: /* KNOWLEDGE FUSION IN § 3.3.2 */
13: $\hat{p} \leftarrow \text{RetrievalBasedPrediction}(\mathcal{RS})$
14: $p \leftarrow \text{InitialPrediction}(x^{\text{test}}, \mathcal{T}, f, g)$
15: $p^{\text{aug}} \leftarrow \text{Fusion}(\hat{p}, p)$
16: **return** $p^{\text{aug}}$;

---

vector whose $j$-th element is $v_j$, and $H(\Gamma) = -\sum_{i,j} \Gamma_{ij} \log \Gamma_{ij}$ is an entropic regularizer for an efficient solution using the Sinkhorn-Knopp algorithm (Cuturi, 2013). $\mathbf{1}_N$ denotes the all-one vector of dimension $N$. Using the optimal solution $\Gamma^*$, the distance between $U$ and $V$ is $\langle \Gamma^*, \mathbf{C} \rangle$.

A key advantage of the OT framework is its flexibility to incorporate a user-defined disparity function to measure the distance between two distributions. In RA-TTA, a novel disparity (i.e., cost) function called *semantic gap* is introduced to calculate the semantic disparity between a test image and a retrieved image. Moreover, we adopt two techniques to enhance the reliability and robustness of the adaptation procedure:

- We reuse the augmented images of the test image, previously employed in description-based retrieval. As a result, we have the test image set $\mathcal{A}$ (comprising the test image and its augmented images) and the retrieved image set $\mathcal{R}_{\mathcal{G}_c}$ that is related to the selected features of class $c$.
- We represent the importance of each image within these sets using alignment scores because the importance of individual images may vary. This importance is normalized, allowing the two sets to be interpreted as probability distributions.

Thus, calculating the distance between the test image and the retrieved images is transformed into calculating the distance between two distributions using our novel disparity function, *semantic gap*. Consequently, the optimal transport framework is adopted in this context.

# D    FURTHER DETAILS FOR EVALUATION

## D.1    DETAILS OF EVALUATION DATASETS

In Table 5, we provide detailed statistics on the evaluation datasets. We use the dataset configuration of CoOp (Zhou et al., 2022) except for Caltech256 and RESISC45, which CoOp does not handle. We use a test split from SuS-X (Udandarao et al., 2023) for Caltech256, and we adopt a test split from (Neumann et al., 2020) for RESISC45.

## D.2    DETAILS OF EXTERNAL IMAGE DATABASE CONSTRUCTION

To construct a web-scale external *image* database, we explore the text captions of LAION2B (Schuhmann et al., 2022) and download the images whose captions contain the target class name, such as "Chevrolet Impala." Specifically, the SQLite full-text search (FTS) table of the text captions in LAION2B is first built, which enables efficient text search for the captions. Then, the class names of downstream tasks are extracted from the definition of downstream tasks. The class names are queried to the FTS table, and the table returns the indexes of the captions that contain the

Table 5: Detailed statistics of datasets used in evaluations.

| Dataset | # Classes | # Test samples | Remark |
|---|---|---|---|
| ImageNet | 1,000 | 50,000 | WordNet categories classification |
| Flowers102 | 102 | 2,463 | Fine-grained flowers classification |
| DTD | 47 | 1,692 | Textural classification |
| Oxford Pets | 37 | 3,669 | Fine-grained pets classification |
| Stanford Cars | 196 | 8,041 | Fine-grained car classification |
| UCF101 | 101 | 3,783 | Human action classification |
| Caltech101 | 100 | 2,465 | Object classification |
| Food101 | 101 | 30,300 | Fine-grained food classification |
| Sun397 | 397 | 19,850 | Scene classification |
| FGVCAircraft | 100 | 3,333 | Fine-grained aircraft classification |
| RESISC45 | 45 | 6,300 | Remote sensing image scene classification |
| Caltech256 | 256 | 2465 | Object classification |
| CUB200 | 200 | 5794 | Fine-grained birds classification |
| ImageNet-V2 | 1,000 | 10,000 | ImageNet variant of temporal shift |
| ImageNet-Sketch | 1,000 | 50,889 | ImageNet variant of sketches |
| ImageNet-Adversarial | 200 | 7,500 | ImageNet variant of adversarial samples |
| ImageNet-Rendition | 200 | 30,000 | ImageNet variant of artistic renditions |

queried class names. Finally, the corresponding images to the returned indexes are downloaded. The database of each downstream task is constructed separately, and the database size for each downstream task is reported in Table 6. Because the database is constructed using the class name, the relevance of the database for each dataset to the corresponding task could be ensured (Wallingford et al., 2023; Parashar et al., 2024).

Table 6: Detailed statistics of external image databases for the evaluation datasets.

| Dataset | $D$ |
|---|---|
| ImageNet | 10,695,986 |
| Flowers102 | 2,128,144 |
| DTD | 452,735 |
| Oxford Pets | 1,328,173 |
| Stanford Cars | 374,413 |
| UCF101 | 1,795,826 |
| Caltech101 | 2,741,240 |
| Food101 | 2,237,734 |
| Sun397 | 6,267,408 |
| FGVCAircraft | 336,551 |
| RESISC45 | 1,962,533 |
| Caltech256 | 7,015,883 |
| CUB200 | 644,720 |

The LAION2B consists of 2B image-caption pairs from the Web; thus, the downloaded images contain the diversity of the Web. At the same time, the text captions of LAION2B are very noisy, and, as shown in Figure 7, the downloaded image could be irrelevant to the queried class name even if the caption contains the queried name, reflecting the Web's noisy nature. Therefore, the constructed databases preserve the nature of a web-scale image database.

## D.3 DETAILS OF THE COMPARED METHODS

We evaluate ten methods to compare with our proposed RA-TTA. The implementation details for the compared methods as follows.

- **CLIP** (Radford et al., 2021): A basic prompt "a photo of a/an <Class>" is used for the zero-shot classification.[3]

- **Ensemble** (Radford et al., 2021): The ensemble of 80 hand-crafted prompts is used as a text prompt.

---

[3] https://github.com/openai/CLIP

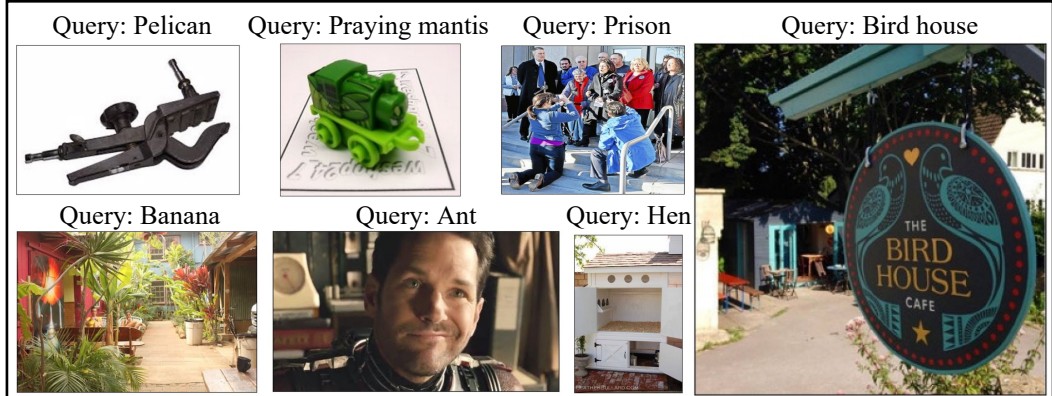

Figure 7: Examples of the downloaded *noisy* images with the corresponding queried class name from ImageNet.

- **TPT** (Shu et al., 2022): The official code[4] released by the authors is used, and their default hyperparameters are adopted.

- **C-TPT** (Yoon et al., 2024): The official code[5] released by the authors is used, and their default hyperparameters are adopted.

- **RLCF** (Zhao et al., 2024b): The official code[6] released by the authors is used, and their default hyperparameters are adopted. The reward model is same with the main model.

- **VisDesc** (Menon & Vondrick, 2022): The official code[7] released by the authors is used. The default templates for generating text descriptions are adopted.

- **WaffleCLIP** (Roth et al., 2023): The official code[8] released by the authors is used. The default templates for generating text descriptions are adopted.

- **CuPL** (Pratt et al., 2023): The official code[9] released by the authors is used. As remarked in Appendix D.4, we slightly modified the default templates for generating text descriptions.

- **SuS-X-LC** (Udandarao et al., 2023): The official code[10] released by the authors is used. Because the validation dataset is unavailable in TTA, hyperparameter tuning for each dataset is prohibited. Thus, the same hyperparameters are used across all datasets, which are reported to function well in the original paper. For the fair comparison between the retrieval-based methods, including SuS-X-LC, Neural Priming, and RA-TTA, the image database introduced in Appendix D.2 is used for the retrieval.

- **Neural Priming** (Wallingford et al., 2023): The official code[11] released by the authors is used, and their default hyperparameters are adopted. As with SuS-X-LC, the image database introduced in Appendix D.2 is used for the retrieval.

## D.4 LLM TEMPLATES FOR GENERATING TEXT DESCRIPTIONS

In Table 7, we present the templates for GPT-3.5 Turbo (Ouyang et al., 2022) used to generate text descriptions for each dataset. The templates are based on those from CuPL (Pratt et al., 2023), but we slightly modified them for GPT-3.5 Turbo. Since the CuPL templates were initially designed for *text-davinci-002*, which is based on GPT-3 and has now been deprecated, some of these templates do not function as expected with GPT-3.5 turbo. For instance, when prompted with the original CuPL template, "Describe an image from the internet of a goldfish," GPT-3.5 turbo responds, "I

---

[4]https://github.com/azshue/TPT
[5]https://github.com/hee-suk-yoon/C-TPT
[6]https://github.com/mzhaoshuai/RLCF
[7]https://github.com/sachit-menon/classify_by_description_release
[8]https://github.com/ExplainableML/WaffleCLIP
[9]https://github.com/sarahpratt/CuPL
[10]https://github.com/vishaal27/SuS-X
[11]https://github.com/RAIVNLab/neural-priming

can't access the internet to see the image of a goldfish." Therefore, we carefully revised only those templates that do not function properly with GPT-3.5 Turbo to ensure they work as intended.

Table 7: CuPL style hand-written templates for extracting text descriptions from LLMs.

| Dataset | CuPL style templates |
|---|---|
| ImageNet | - Describe a photo of {} {}.
- Describe what {} {} looks like.
- How can you identify {} {}?
- What does {} {} look like?
- Describe a photo of {} {} with distinctive visual features.
- Describe the shapes and structures of .
- Describe the color and patterns of .
- Describe the overall appereance of .
- A caption of an image of {} {}: |
| Flowers102 | - What does {} {} flower look like.
- Describe the appearance of {} {}.
- Visually describe {} {}, a type of flower.
- A caption of an image of {} {}. |
| DTD | - What does {} {} material look like? Please explain the characteristics and provide examples of where it is typically found.
- What does {} {} surface look like? Please explain the characteristics and provide examples of where it is typically found.
- What does {} {} texture look like? Please explain the characteristics and provide examples of where it is typically found.
- What does {} {} object look like? Please explain the characteristics and provide examples of where it is typically found.
- What does {} {} thing look like? Please explain the characteristics and provide examples of where it is typically found.
- What does {} {} pattern look like? Please explain the characteristics and provide examples of where it is typically found. |
| Oxford Pets | - Describe what {} {} pet looks like.
- Visually describe {} {}, a type of pet. |
| Stanford Cars | - How can you identify {} {}.
- Description of {} {}, a type of car.
- A caption of a photo of {} {}:
- What are the primary characteristics of {} {}?
- Description of the exterior of {} {}.
- What are the identifying characteristics of {} {}, a type of car?
- Describe an image of {} {}.
- What does {} {} look like?
- Describe what {} {}, a type of car, looks like. |
| UCF101 | - What does a person doing {} look like.
- Describe the process of {}.
- How does a person {}. |
| Caltech101 | - Describe what {} {} looks like.
- What does  look like.
- Describe the shapes and structures of .
- Describe a photo of {} {} with distinctive visual features. |
| Food101 | - Visually describe {} {}.
- Describe what {} {} looks like.
- Describe a photo of {} {} with distinctive visual features.
- How can you tell that the food in a photo is {} {}? |
| SUN397 | - Describe what {} {} looks like.
- How can you identify {} {}?
- Describe a photo of {} {} with distinctive visual features. |
| FGVC Aircraft | - Describe {} {} aircraft. |
| RESISC45 | - Describe a satellite photo of {} {}.
- Describe the satellite photo of {} {}.
- Describe an aerial photo of {} {}. |
| Caltech256 | - Describe what {} {} looks like.
- Describe a photo of {} {} with distinctive visual features. |
| CUB200 | - Describe what {} {}, a species of bird, looks like.
- What does {} {} look like.
- Visually describe {} {}, a type of bird.
- A caption of an image of {} {}, a type of bird.
- Describe the appearance of {} {}.
- What are the prominent features to identify {} {} bird. |
| ImageNet rendition | - Describe an art drawing of {} {}.
- Describe artwork showing {} {}.
- Describe a cartoon {} {}.
- Describe an origami of {} {}.
- Describe a deviant art photo depicting {} {}.
- Describe an embroidery of {} {}.
- Describe a graffiti art showing {} {}.
- Describe a painting of {} {}.
- Describe a sculpture of {} {}.
- Describe a black and white sketch of {} {}.
- Describe a toy {} {}.
- Describe a video game of {} {}. |
| ImageNet sketch | - Describe how a black and white sketch of {} {} looks like.
- Describe a black and white sketch of {} {}. |

# E  ADDITIONAL EVALUATION RESULTS

## E.1  MAIN RESULTS WITH STATISTICAL SIGNIFICANCE

In Tables 8 and 9, we report the standard deviation omitted in Tables 1 and 2, where the value after "$\pm$" represents the standard deviation over *five* different runs. Overall, RA-TTA demonstrates robust performance, exhibiting low variability across most datasets.

Table 8: Statistical significance for the results in Table 1.

| | IN-1k | Flowers102 | DTD | Oxford Pets | Stanford Cars | UCF101 | Caltech101 | Food101 | SUN397 | FGVC Aircraft | RESISC45 | Caltech256 | CUB200 | Avg. (13) |
|---|---|---|---|---|---|---|---|---|---|---|---|---|---|---|
| TPT | 69.08 ($\pm$0.08) | 69.18 ($\pm$0.12) | 47.04 ($\pm$0.23) | 87.44 ($\pm$0.09) | 66.55 ($\pm$0.06) | 68.04 ($\pm$0.33) | 93.79 ($\pm$0.08) | 86.34 ($\pm$0.15) | 65.32 ($\pm$0.08) | 23.31 ($\pm$0.21) | 56.84 ($\pm$0.19) | 85.37 ($\pm$0.17) | 60.11 ($\pm$0.16) | 67.57 ($\pm$0.15) |
| C-TPT | 68.32 ($\pm$0.07) | 69.43 ($\pm$0.16) | 45.27 ($\pm$0.14) | 88.25 ($\pm$0.13) | 65.48 ($\pm$0.15) | 65.50 ($\pm$0.20) | 93.39 ($\pm$0.22) | 84.95 ($\pm$0.21) | 64.55 ($\pm$0.22) | 24.39 ($\pm$0.17) | 56.02 ($\pm$0.09) | 85.25 ($\pm$0.11) | 58.84 ($\pm$0.30) | 66.90 ($\pm$0.17) |
| RLCF | 68.61 ($\pm$0.24) | 67.72 ($\pm$0.31) | 46.40 ($\pm$0.23) | 86.73 ($\pm$0.20) | 66.51 ($\pm$0.30) | 66.98 ($\pm$0.23) | 93.83 ($\pm$0.28) | 86.09 ($\pm$0.28) | 64.92 ($\pm$0.22) | 23.43 ($\pm$0.27) | 56.89 ($\pm$0.18) | 85.18 ($\pm$0.11) | 57.91 ($\pm$0.31) | 67.02 ($\pm$0.23) |
| **RA-TTA (Ours)** | **70.58** ($\pm$0.04) | **78.65** ($\pm$0.10) | **60.98** ($\pm$0.17) | **92.78** ($\pm$0.09) | **70.11** ($\pm$0.10) | **73.28** ($\pm$0.27) | **94.84** ($\pm$0.08) | **87.10** ($\pm$0.04) | **70.38** ($\pm$0.05) | **32.34** ($\pm$0.22) | **66.95** ($\pm$0.06) | **89.50** ($\pm$0.07) | **62.73** ($\pm$0.17) | **73.09** ($\pm$0.12) |

Table 9: Statistical significance for the results in Table 2.

| | IN-A | IN-V2 | IN-R | IN-K | Avg. (4) |
|---|---|---|---|---|---|
| TPT | 55.78 ($\pm$0.17) | 63.48 ($\pm$0.14) | 78.98 ($\pm$0.10) | 47.95 ($\pm$0.12) | 60.77 ($\pm$0.13) |
| C-TPT | 50.28 ($\pm$0.22) | 62.47 ($\pm$0.28) | 75.68 ($\pm$0.07) | 47.42 ($\pm$0.13) | 58.96 ($\pm$0.18) |
| RLCF | 56.52 ($\pm$0.18) | 63.37 ($\pm$0.22) | 77.04 ($\pm$0.05) | 48.09 ($\pm$0.14) | 61.26 ($\pm$0.15) |
| **RA-TTA (Ours)** | **59.21** ($\pm$0.26) | **64.16** ($\pm$0.15) | **79.68** ($\pm$0.04) | **50.83** ($\pm$0.04) | **63.47** ($\pm$0.12) |

## E.2  RESULTS ON ABLATION STUDY

We present the implementation details for the variants of RA-TTA. For the image-to-image similarity search of Var. 1, we use simple cosine similarity between image embeddings and retrieve $K_S$ external images for a test image. To replace the semantic gap computation for Var. 2, we adopt simple cosine similarity between image embeddings. To realize Var. 3, we just use uniform weights vectors for $\mathcal{U}$ and $\mathcal{V}$.

In Table 10, we present results on ablation study across the standard transfer learning benchmarks, and the similar trends in Section 4.3 are observed, where description-based approaches, including Var. 3 and RA-TTA, outperform other variants.

Table 10: Results on ablation study for test-time adaptation (or transfer learning).

| | IN-1k | Flowers102 | DTD | Oxford pets | Stanford cars | UCF101 | Caltech101 | Food101 | SUN397 | FGVC aircraft | RESISC45 | Caltech256 | CUB200 | Avg. (13) |
|---|---|---|---|---|---|---|---|---|---|---|---|---|---|---|
| Var. 1 | 68.62 | 77.92 | 56.67 | 91.22 | 67.81 | 67.44 | 93.58 | 86.41 | 68.62 | 29.39 | 62.92 | 88.43 | 60.78 | 70.75 |
| Var. 2 | 68.81 | 78.81 | 57.17 | 91.93 | 68.71 | 69.02 | 94.05 | 86.93 | 69.41 | 30.91 | 65.01 | 88.61 | 61.27 | 71.59 |
| Var. 3 | 70.45 | **79.31** | 59.16 | 91.94 | 69.95 | 73.14 | 94.35 | **87.15** | 69.86 | 31.96 | 65.67 | 88.82 | **62.92** | 72.67 |
| **RA-TTA (Ours)** | **70.58** | 78.65 | **60.98** | **92.78** | **70.11** | **73.28** | **94.84** | 87.10 | **70.38** | **32.34** | **66.95** | **89.50** | 62.73 | **73.09** |

## E.3  EFFECTS OF DIFFERENT LLMs FOR GENERATING TEXT DESCRIPTIONS

The description-based retrieval relies on the LLMs to generate detailed text descriptions. Thus, to analyze how different LLMs affect description quality and final adaptation performance, text descriptions generated by Claude-Sonnet are newly considered for this analysis. We compare the

performance of CuPL (Pratt et al., 2023) to investigate the quality of text descriptions, as CuPL can achieve better performance with improved descriptions.

As shown in the first two rows of Table 11, CuPL with the descriptions of GPT works better than CuPL with those of Claude, implying that the text descriptions generated by GPT are better than those generated by Claude. Thus, different LLMs can affect description quality even with the same

Table 11: Performance when using text descriptions generated by ChatGPT and Claude3. We report the top-1 accuracy (%) on the FGVC aircraft, Stanford cars, and RESISC45 datasets. The employed LLMs are indicated with the method name.

|  | FGVC aircraft | Stanford cars | RESISC45 | Avg. (3) |
|---|---|---|---|---|
| CuPL+Claude3 | 28.05 | 66.38 | 57.33 | 50.59 |
| CuPL+ChatGPT | 28.89 | 66.92 | 65.17 | 53.66 |
| **RA-TTA**+Claude3 | 30.93 | 68.82 | 61.92 | 53.89 |
| **RA-TTA**+ChatGPT | **32.34** | **70.11** | **66.95** | **56.47** |

generation prompts. Also, the last two rows of Table 11 show that RA-TTA with the descriptions of GPT outperforms RA-TTA with those of Claude, implying that the better descriptions facilitate the adaptation of RA-TTA more effectively by improving the relevance and quality of retrieved images. Notably, RA-TTA with the text descriptions of Claude outperforms the performance of the corresponding CuPL (i.e., CuPL+Claude3 vs. RA-TTA+Claude3), demonstrating that RA-TTA effectively leverages external knowledge even when the quality of the description is not very good.

### E.4 IMPACT OF DATABASE SIZE ON ADAPTATION PERFORMANCE

Regarding the database of external images, we analyze the impact of its size on adaptation performance—adaptation accuracy, storage costs, and inference time. For this analysis, we use random sampling to build a subset of the original database. As shown in Table 12, accuracy improves alongside storage costs as the database size increases. This result is expected because the likelihood of including informative samples for adaptation increases with a larger database, albeit at the expense of increased storage costs. Regarding inference time, thanks to efficient search engines, such as FAISS (Johnson et al., 2021), the increase in latency remains almost negligible, as shown in the last row of Table 12.

Table 12: Effects of the database size on accuracy, storage costs, and inference time of RA-TTA. We report the top-1 accuracy (%), storage costs (MB), and average GPU inference time per sample (s/sample) measured on a single RTX4090 for the Stanford cars dataset. The percentage represents the proportion of the sampled database to the original database. The values in the parentheses indicate the number of images in a database.

|  | 5% (13,720) | 10% (37,441) | 20% (74,882) | 40% (149,765) | 60% (224,647) | 80% (299,530) | Original (374,413) |
|---|---|---|---|---|---|---|---|
| Accuracy | 68.12 | 68.81 | 69.01 | 69.81 | 69.96 | 70.08 | 70.11 |
| Storage costs (MB) | 18.28 | 36.56 | 73.13 | 146.25 | 219.38 | 292.51 | 365.64 |
| Inference time (s/sample) | 0.106 | 0.106 | 0.107 | 0.111 | 0.113 | 0.113 | 0.117 |

### E.5 RESULTS ON SENSITIVITY ANALYSIS

We present hyperparameter sensitivity analyses on the standard transfer learning datasets, and the similar trends in Section 4.4 are observed as shown in Figures 8–20.

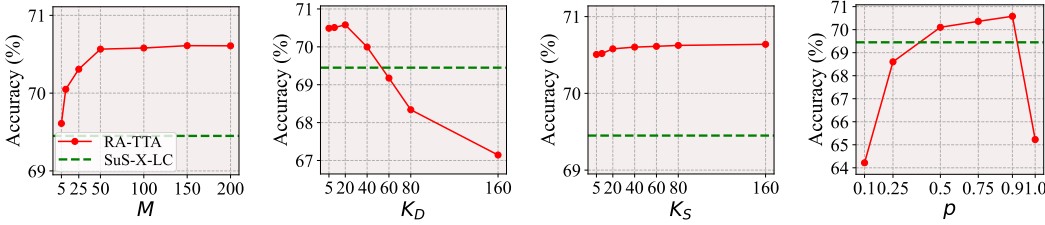

(a) Augmentation size.   (b) # of selected descriptions.   (c) # of retrieved images.   (d) Score percentile.

Figure 8: Effects of the four hyperparameters on the accuracy of ImageNet-1k.

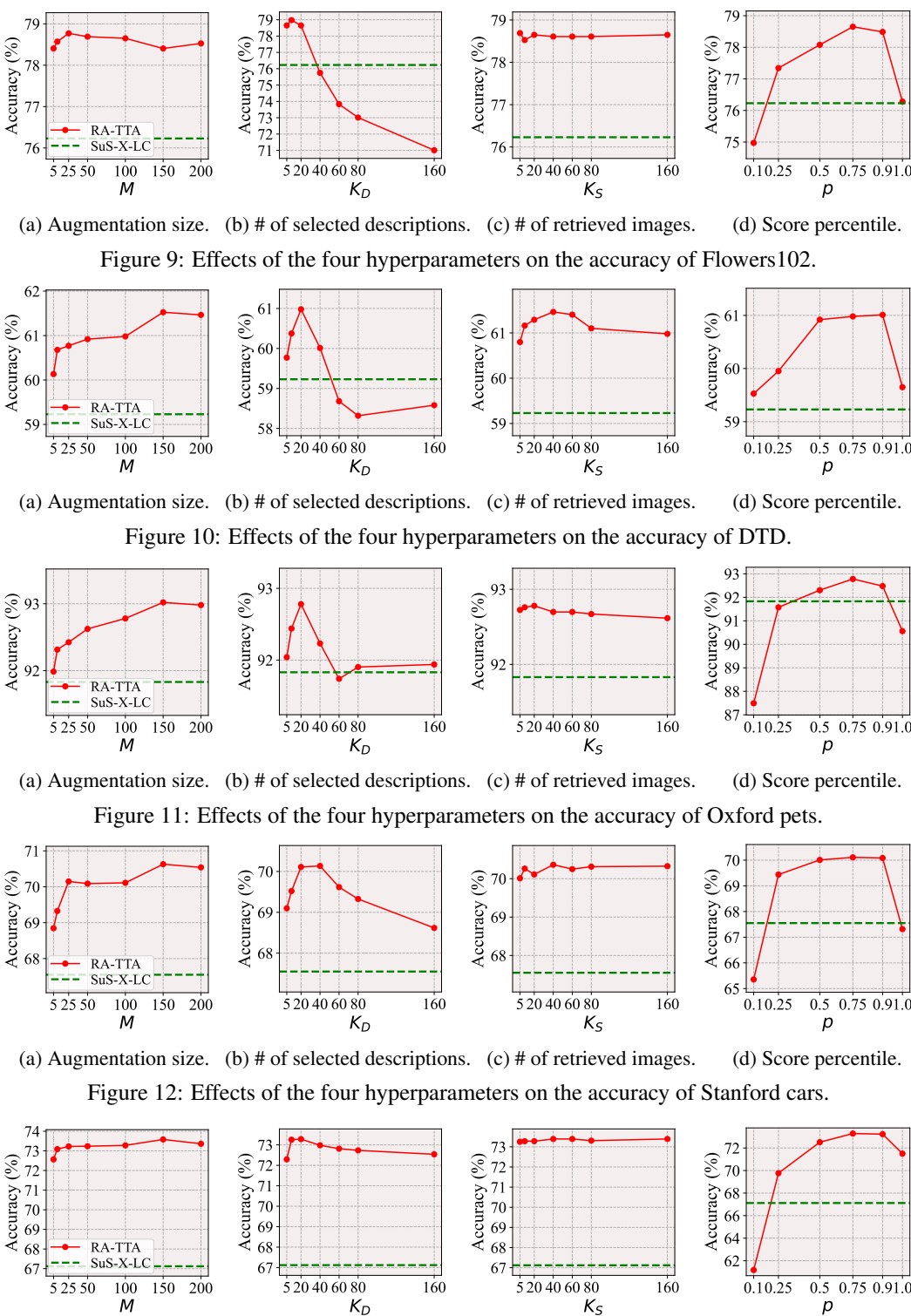

(a) Augmentation size.    (b) # of selected descriptions.    (c) # of retrieved images.    (d) Score percentile.

Figure 9: Effects of the four hyperparameters on the accuracy of Flowers102.

(a) Augmentation size.    (b) # of selected descriptions.    (c) # of retrieved images.    (d) Score percentile.

Figure 10: Effects of the four hyperparameters on the accuracy of DTD.

(a) Augmentation size.    (b) # of selected descriptions.    (c) # of retrieved images.    (d) Score percentile.

Figure 11: Effects of the four hyperparameters on the accuracy of Oxford pets.

(a) Augmentation size.    (b) # of selected descriptions.    (c) # of retrieved images.    (d) Score percentile.

Figure 12: Effects of the four hyperparameters on the accuracy of Stanford cars.

(a) Augmentation size.    (b) # of selected descriptions.    (c) # of retrieved images.    (d) Score percentile.

Figure 13: Effects of the four hyperparameters on the accuracy of UCF101.

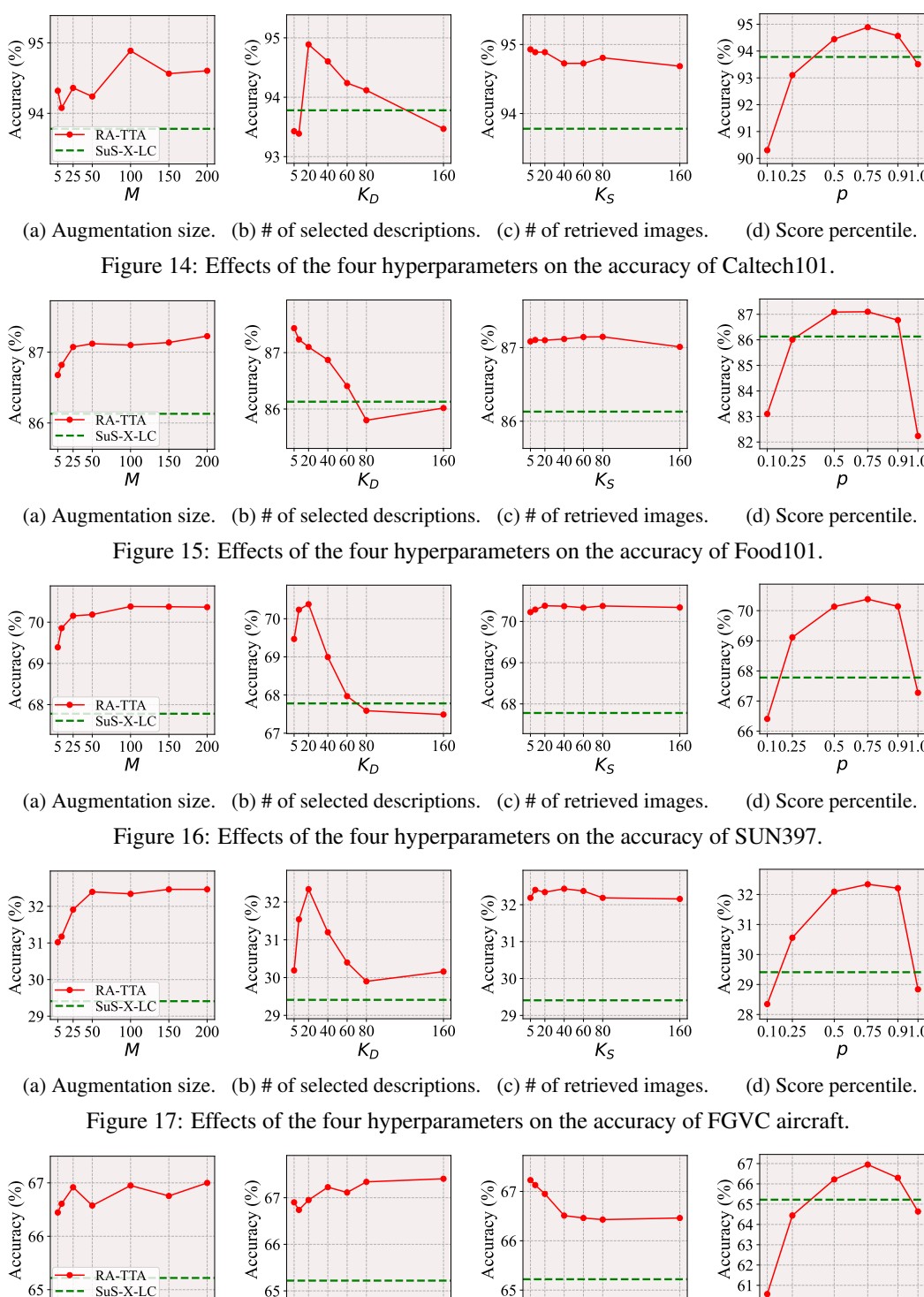

(a) Augmentation size.  (b) # of selected descriptions.  (c) # of retrieved images.  (d) Score percentile.

Figure 14: Effects of the four hyperparameters on the accuracy of Caltech101.

(a) Augmentation size.  (b) # of selected descriptions.  (c) # of retrieved images.  (d) Score percentile.

Figure 15: Effects of the four hyperparameters on the accuracy of Food101.

(a) Augmentation size.  (b) # of selected descriptions.  (c) # of retrieved images.  (d) Score percentile.

Figure 16: Effects of the four hyperparameters on the accuracy of SUN397.

(a) Augmentation size.  (b) # of selected descriptions.  (c) # of retrieved images.  (d) Score percentile.

Figure 17: Effects of the four hyperparameters on the accuracy of FGVC aircraft.

(a) Augmentation size.  (b) # of selected descriptions.  (c) # of retrieved images.  (d) Score percentile.

Figure 18: Effects of the four hyperparameters on the accuracy of RESISC45.

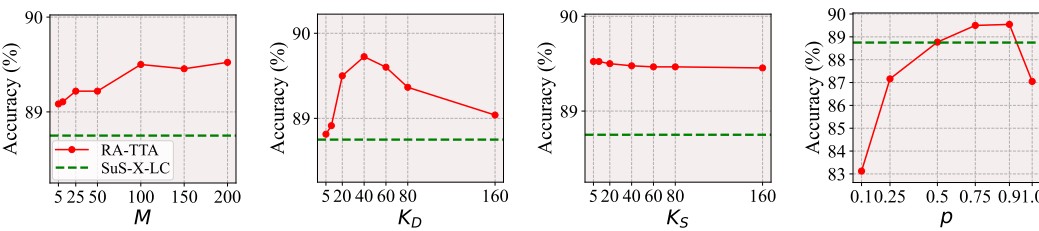

(a) Augmentation size.    (b) # of selected descriptions.    (c) # of retrieved images.    (d) Score percentile.

Figure 19: Effects of the four hyperparameters on the accuracy of Caltech256.

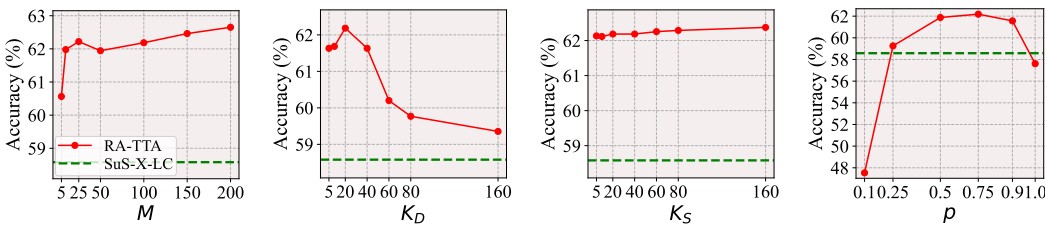

(a) Augmentation size.    (b) # of selected descriptions.    (c) # of retrieved images.    (d) Score percentile.

Figure 20: Effects of the four hyperparameters on the accuracy of CUB200.

### E.6  GPU MEMORY OVERHEAD FOR INFERENCE

RA-TTA does not require backpropagation (i.e., training-free), which contributes to the memory efficiency of RA-TTA. As shown in Table 13, it results in lower GPU peak memory compared to TPT which uses backpropagation to update learnable prompts.

Table 13: GPU peak memory for TPT and RA-TTA. We report the GPU peak memory (MB) on the FGVC aircraft, Stanford cars, and RESISC45 datasets.

|  | FGVC aircraft | Stanford cars | RESISC45 | Avg. (3) |
|---|---|---|---|---|
| TPT | 2213.53 | 3718.75 | 1358.73 | 2430.33 |
| **RA-TTA (Ours)** | **905.63** | **984.75** | **911.43** | **933.94** |

### E.7  ADDITIONAL VISUALIZATIONS

In Figure 21, we present illustrative examples of the rejected/selected descriptions in description-based retrieval on the Stanford cars (Krause et al., 2013) and RESISC45 Cheng et al. (2017) datasets. Figure 21 illustrates that the selected descriptions pay attention to a specific piece of information, namely a pivotal feature, whereas the rejected descriptions emphasize irrelevant features. Figure 22 briefly illustrates the workflow of RA-TTA on the Stanford cars and RESISC45 datasets.

| Test image | Rejected text descriptions with **irrelevant** features | Selected text descriptions with **relevant** features |
|---|---|---|
| GT: 2012 Porsche Panamera Sedan | You can identify a 2012 Porsche Panamera Sedan by looking for its distinct body style, four-door design, aggressive **front fascia** and signature **Porsche logo on the front and rear** of the vehicle. | The 2012 Porsche Panamera Sedan features sporty exterior styling, which includes a **sleek, low-slung profile with four doors**, and **wide wheel arches**. It has a distinctive **sloping roofline** and a **four-door layout**, combining the elegance of a coupe. |
| GT: Thermal power plant | A thermal power plant in an aerial image would show large buildings with tall chimneys and **a network of power lines connecting it to the grid**. It also have fuel storage areas, **water reservoirs**, and **roads for transport.** | A thermal power plant in a satellite photo appears as a large industrial complex with prominent structures like **cooling towers, boiler units**, and **smokestacks**. The photo may show **heat emissions in the form of thermal plumes from the cooling towers and steam vents**, as well as energy production. |
| GT: Harbor | A satellite photo of a harbor typically shows ships and boats docked along piers and quays, with **cranes and containers visible for cargo handling**. The layout of the harbor, such as **breakwaters** can be seen along with surrounding infrastructure. | A satellite photo of a harbor typically shows **a body of water surrounded** by land with **docks, piers, and boats**. The harbor may appear as a dark area within the surrounding land, with **watercraft visible in the water**. |

Figure 21: Examples of the selected and rejected text descriptions in our description-based retrieval on the Stanford cars and RESISC45 datasets. The results of the rejected text descriptions and the selected text descriptions are shown individually. An irrelevant feature is colored yellow in the descriptions, while a relevant feature is colored red.

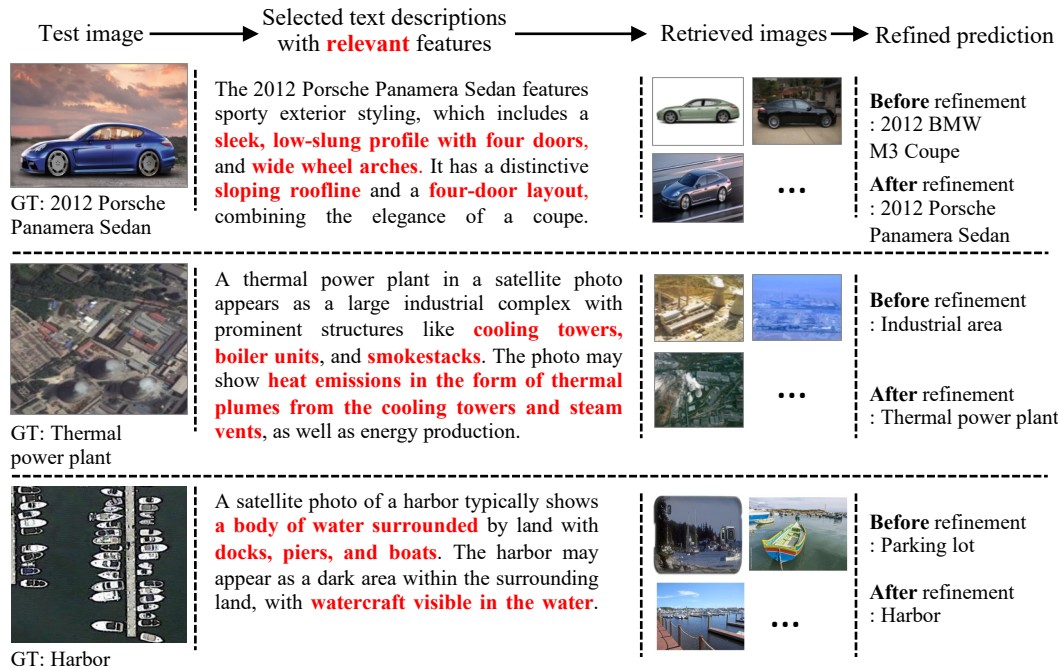

Figure 22: Examples of the brief workflow of RA-TTA on the Stanford cars and RESISC45 datasets. The test image is queried to select text descriptions that include relevant features, and then proper images are retrieved based on those text descriptions. The initial prediction is refined with the help of the retrieved images.

