# OpenReview forum: "RA-TTA: Retrieval-Augmented Test-Time Adaptation for Vision-Language Models"
_ICLR.cc/2025/Conference — ICLR 2025 Poster_

### Official Review · Reviewer_DF2M · 2024-10-28

**Soundness:** 3
**Presentation:** 2
**Contribution:** 3
**Rating:** 6
**Confidence:** 4

**Summary:**

This work focuses on test-time adaptation (TTA) of vision-language models (VLMs) and proposes retrieval-augmented TTA (RA-TTA), aiming to improve the generalization ability of VLMs on downstream datasets. Specifically, RA-TTA incorporates external visual information by retrieving web-scale images with fine-grained text descriptions, which are then used to augment VLM predictions at test time. Unlike prior test-time adaptation methods that rely solely on the pre-trained model's internal knowledge, RA-TTA enhances test-time adaptation with retrieved images, consistently improving prediction accuracy across 17 datasets.

**Strengths:**

1. Introducing retrieval augmentation into test-time adaptation of VLMs is novel. Prior methods generally augment test images using simple augmentation strategies (e.g., crop, resize), while this work exploits the external knowledge form additional image data bases to augment each test image for better model predictions. Additionally, the proposed description-based retrieval could effectively leverages the bi-modality of VLMs, leading to more relevant retrieval results.
2. The experiments conducted in this work are extensive. The authors benchmark the proposed RA-TTA with the SOTA methods on 17 diverse datasets, and provide comprehensive analysis experiments, including ablation studies and parameter sensitivity analyses.

**Weaknesses:**

1. RA-TTA highly relies on Large Language Models (LLMs) to generate fine-grained text descriptions for image retrieval. Therefore, the authors should discuss how different LLMs might vary in their effectiveness at generating useful descriptions, and how this impacts retrieval relevance and quality.
2. The authors construct different data base for each downstream dataset by sampling form a large dataset such as LAION2B. The sampled database size may impact both computational cost and retrieval quality, for example, a larger sampled database size may bring better retrieval quality but may also increase latency and storage costs. Therefore, an analysis of how the sampled database size affects adaptation performance should be provided.
3. Ensuring semantic diversity in text descriptions for each class name is crucial in RA-TTA. How is this diversity ensured during the generation process?
4. Compared  with prior studies such as TPT, the proposed RA-TTA involves additional database construction process and retrieval process which may bring more computation cost and inference time. Therefore, the computational and storage overhead compared to TPT should be provided.

**Questions:**

Please refer to the weakness section.

---

> ### Author Response · Authors · 2024-11-22
> **Author's Response to Reviewer DF2M (1/2)**
>
> We sincerely thank the reviewer for your insightful comments. Below, we are more than willing to clarify the reviewer’s concerns one by one.
>
> ```Q1: The authors should discuss how different LLMs might vary in their effectiveness at generating useful descriptions, and how this impacts retrieval relevance and quality.```
>
> We appreciate the reviewer for providing a valuable discussion point. We conducted additional experiments using text descriptions generated by Claude3-Sonnet for this analysis.
> - We also compared the performance of CuPL [1] to investigate the quality of text descriptions because CuPL can achieve better performance with improved descriptions. As shown in the first two rows of Table R1, CuPL with the descriptions of GPT works better than CuPL with those of Claude, implying that the text descriptions generated by GPT are better than those of Claude. Thus, we can conclude that different LLMs can affect description quality even with the same generation prompts.
> - Also, the last two rows of Table R1 show that RA-TTA with the descriptions of GPT outperformed RA-TTA with those of Claude. This indicates that the better descriptions facilitate the adaptation of RA-TTA more effectively by improving the relevance and quality of retrieved images.
> - Notably, RA-TTA with the text descriptions of Claude outperformed the performance of corresponding CuPL (i.e., CuPL+Claude3 vs. RA-TTA+Claude3), demonstrating that RA-TTA effectively leverages external knowledge, even when the quality of the description is not very good.
>
> We have added this analysis in **Appendix D.5** of the revised manuscript.
>
> Table R1: Performance when using descriptions generated by ChatGPT and Claude3. We report the top-1 accuracy (%) on the FGVC aircraft, Stanford cars, and RESISC45 datasets. The employed LLMs are indicated with the method name. The best results are in **bold**. The "Avg." column indicates the average accuracy of 3 datasets.
> | **Method+LLM** | FGVC aircraft    | Stanford cars | RESISC45 | Avg. (3)     |
> |----------- |:------------------:|:---------------:|:---------:|:---------:|
> | **CuPL+Claude3**    |  28.05  |    66.38      |   57.33  |   50.59 |
> | **CuPL+ChatGPT**    |  28.89  |    66.78      |   63.98  |   53.22 |
> | **RA-TTA+Claude3**  |  30.93  |    68.82      |   61.92  |   53.80 |
> | **RA-TTA+ChatGPT**  |  **31.51**  |    **69.43**      |   **65.14**  | **55.36** |
>
> ```Q2: An analysis of how the sampled database size affects adaptation performance should be provided.```
>
> We are grateful to the reviewer for bringing up this crucial discussion point. With respect to the database size, we analyzed the adaptation performance, including adaptation accuracy, storage costs, and inference time. Note that we used random sampling to build the subset of the original database.
>
> - In the first two rows of Table R2, as the database size increases, accuracy improves alongside storage costs. This result is expected because the likelihood of including informative samples for adaptation increases with a larger database, albeit at the expense of increased storage costs.
> - Regarding inference time, thanks to efficient search engines, such as FAISS, the increase in latency remains negligible, as shown in the last row of Table R2.
>
> Additionally, because web-scale datasets often contain noises, we can improve the adaptation accuracy even with a smaller database (i.e., less storage costs) by conducting sampling with advanced filtering tools.
>
> We have added this analysis to **Appendix D.6** of the revised manuscript.
>
> Table R2: Effects of the database size on accuracy, storage costs, and inference time of RA-TTA. We report the top-1 accuracy (%), storage costs (MB), and average GPU inference time per sample (s/sample) measured on a single RTX4090 for the Stanford cars dataset. The percentage represents the proportion of the sampled database to the original database. The values in the parentheses indicate the number of images in a database.
> | **Metric**          | 5% (18,720) | 10% (37,441) | 20% (74,882) | 40% (149,765) | 60% (224,647) | 80% (299,530) | Original (374,413) |
> |---------------------|:-----:|:-----:|:-----:|:-----:|:-----:|:-----:|:-----:|
> | **Accuracy (%)**        | 67.63 | 68.62 | 68.84 | 69.03 | 69.21 | 69.24 | 69.43 |
> | **Storage cost (MB)**    | 18.28 | 36.56 | 73.13 | 146.25| 219.38| 292.51| 365.64|
> | **Inference time (s/sample)**  | 0.106 | 0.106 | 0.107 | 0.111 | 0.113 | 0.113 | 0.117 |

---

> ### Author Response · Authors · 2024-11-22
> **Author's Response to Reviewer DF2M (2/2)**
>
> ```Q3: How is the diversity in text descriptions ensured during the generation process.```
>
> To ensure semantic diversity in text descriptions, we utilized multiple prompts that have different semantics when querying LLMs. Furthermore, we employed a high temperature setting of 0.99 and queried each prompt multiple times.
>
> ```Q4: Therefore, the computational and storage overhead compared to TPT should be provided.```
>
> We appreciate the reviewer for pointing out this important issue.
>
> The cost of constructing databases is inevitable for RAG-based approaches. However, the database is built **only once offline** prior to test time, and the initial cost of database construction is not significantly high because leveraging an advanced tool like FAISS typically used for RAG-based approaches can efficiently manage indexing and storage, minimizing the overhead.
>
> In the context of test-time adaptation, the efficiency of adaptation at test time is indeed a critical factor to consider. In contrast to previous TTA methods, RA-TTA needs to retrieve proper external images for each test image, potentially posing a significant bottleneck. However, the efficiency of RA-TTA is **comparable** to the representative test-time adaptation method, TPT, as shown in Table R3, where we compared the GPU inference time per sample (i.e., s/sample) for TPT [2] and RA-TTA. This is because FAISS, an advanced and efficient search engine typically used for RAG-based approaches, enables **fast nearest-neighbor search**. Moreover, our description-based adaptation does not require backpropagation (i.e., training-free), which also contributes to the efficient inference of RA-TTA.
>
> Table R3: Inference time for TPT and the proposed RA-TTA. We report the average GPU inference time per sample (s/sample) measured on a single RTX4090 for the FGVC aircraft, Stanford cars, and RESISC45 datasets. The "Avg." column indicates the average inference time of 3 datasets.
> | **Method** |   FGVC aircraft (s/sample)  | Stanford cars (s/sample)  |  RESISC45 (s/sample) | Avg. (3) |
> |------------|:----------------:|:---------------:|:----------:|:----------:|
> | **TPT** [2]    |     **0.103**    |    0.155     |  **0.95** | 0.118|
> | **RA-TTA**     |     0.113        |    **0.117**    |    0.121   | **0.117**|
>
> Furthermore, RA-TTA does not need a backpropagation for adaptation, resulting in **low GPU peak memory** compared to TPT that updates learnable prompts using backpropagation as shown in Table R4.
>
> We have added this efficiency analysis in **Appendix D.4 of** the revised manuscript.
>
> Table R4: GPU peak memory for TPT and the proposed RA-TTA. We report the GPU peak memory (MB) on the FGVC aircraft, Stanford cars, and RESISC45 datasets. The "Avg." column indicates the average GPU peak meamory of 3 datasets.
> | **Method** | FGVC aircraft (MB)    | Stanford cars (MB)  | RESISC45 (MB)  | Avg. (3) |
> |------------|:----------------:|:---------------:|:----------:|:----------:|
> | **TPT** [2]    |     2213.53      |  3718.75        |  1358.73   | 2430.33|
> | **RA-TTA** |     **905.63**   |   **984.75**    |  **911.43**| **933.94** |
>
> Once again, we appreciate the reviewer highlighting valuable discussion points in our study. We hope our response has clarified the contents of the paper, and we are more than willing to engage in further discussion regarding any aspects that remain unclear.
>
> ------------------------------------------------
> [1] "What Does a Platypus Look Like? Generating Customized Prompts for Zero-shot Image Classification.", In ICCV, 2023
> [2] "Test-Time Prompt Tuning for Zero-Shot Generalization in Vision-Language Models.", In NeurIPS, 2022

---

> ### Author Response · Authors · 2024-11-26
> **Kindly Follow-Up: Reviewer DF2M**
>
> Dear Reviewer: DF2M,
>
> Thank you for your review and valuable feedback, which have greatly contributed to enhancing the quality of our paper. This is a kind reminder that we only have one and a half days (about 36 hours) before the author/reviewer discussion period ends. We submitted an answer addressing your questions, including the usage of a different LLM, an analysis of the effects of database size, and computation and storage overhead.  We hope that our response has provided sufficient clarification. Please let us know if you have any further inquiries or require additional information.
>
> Best regards,
> Authors.

---

> > ### Comment · Reviewer_DF2M · 2024-11-26
> >
> > Thank you for your thoughtful rebuttal. I appreciate the effort you have put into addressing my comments and concerns. After carefully reviewing your responses, I acknowledge the points raised and the improvements made to improve your work.
> > With the newly presented information, I believe my initial evaluation and rating remain consistent with my overall assessment of the paper. Hence, I would keep my original rating.

---

> > > ### Author Response · Authors · 2024-11-26
> > >
> > > We sincerely appreciate your acknowledgment of our responses. Moreover, we are truly grateful for the time and consideration you have invested in reviewing our manuscript.
> > >
> > > If you have any further questions or concerns, please let us know. We are more than willing to address any additional questions or concerns the reviewer might have.
> > >
> > > With warmest regards,
> > > Authors.

---

### Official Review · Reviewer_aU8U · 2024-10-31

**Soundness:** 3
**Presentation:** 3
**Contribution:** 2
**Rating:** 5
**Confidence:** 4

**Summary:**

In vision-language models, existing test-time adaptation methods solely rely on the internal knowledge encoded within the parameters. To complement the limitation of the internal knowledge, this paper proposes retrieval-augment TTA for adapting VLMs to test distribution using external knowledge obtained from a web-scale image database. It uses fine-grained text descriptions for retrieving proper external images and refining VLM’s predictions with the retrieved external images. Extensive evaluations on 17 datasets validate that RA-TTA outperforms the state-of-the-art methods.

**Strengths:**

1. The paper is well-written and easy to follow.
2. This work investigates test-time adaptation in visual language models. Due to the difficulty of updating pre-trained models frequently, TTA on visual language models is a highly researched topic.
3. The proposed method can achieve state-of-the-art results on 17 benchmarks, including standard transfer learning and distribution shift. And it conducts ablation study to validate the effectiveness of proposed modules.

**Weaknesses:**

1.	The biggest issue with the article is its lack of innovation. In fact, the proposed method is very similar to [1], which is not even cited in this paper. [1] firstly retrieves relevant samples from a web-scale database using seed prompts (T2I) or seed images (I2I), then build a K-shot cache by selecting the Top-K similar images per class based on CLIP embeddings, and the final predictions is an ensemble of original predictions and few-shot cache predictions. The basic idea and pipeline are very similar to [1].
2.	In Table2, the proposed method achieves inferior performance than TPT [2] and RLCF [3] in IN-V2 and IN-A. The authors should analyze the reasons behind this phenomenon. Does it indicate that the effectiveness of the method is related to the downstream datasets?
3. The paper lacks clarity in describing certain operations. For instance, in the Equation 8, why is optimal transport used, and what is its purpose?


[1] Understanding Retrieval-Augmented Task Adaptation for Vision-Language Models, ICML 2024.

[2] Test-time Prompt Tuning for Zero-shot Generalization in Vision-Language Models, NeurIPS 2022.

[3] Test-time Adaptation with CLIP Reward for Zero-shot Generalization in Vision-Language Models. In ICLR 2022.

**Questions:**

1. The basic idea and pipeline are very similar to [1]. The differences and advantages should be claimed.

2.  In Equation 8, why is optimal transport used, and what is its purpose?

3. Figure 3 is not very clear in presenting the proposed method, specifically in prototype-based retrieval part.

[1]  Understanding Retrieval-Augmented Task Adaptation for Vision-Language Models, ICML 2024.

---

> ### Author Response · Authors · 2024-11-22
> **Author's Response to Reviewer aU8U (1/2)**
>
> Thanks for your very constructive comments and reasonable concerns, which have helped us improve the paper quality significantly. Below, we would like to provide point-to-point responses to all the raised questions.
>
> ```Q1(W1): The differences and advantages with [1] should be claimed```
>
> The proposed RA-TTA focused on test-time adaptation (TTA), emphasizing the importance of **adaptive** responses to test images provided online. The primary novelty of RA-TTA, compared to [1], lies in its **adaptiveness**. Specifically, RA-TTA adaptively retrieves external images for each test image based on its pivotal features. In contrast, [1] inherently lacks this adaptive retrieval capability, which is an essential requirement for effective TTA.
>
> This lack of adaptiveness of retrieval-based VLM adaptation, including SuS-X [2] and Neural Priming [3], had already been discussed in Section 2.3 of our manuscript, but the citation to [1] was indeed missing. We have now added the citation, which has further strengthened the clarity and completeness of our paper.
>
> To demonstrate the effectiveness of the proposed **adaptive** retrieval strategy, we conducted a performance comparison with [1]. Although [1] proposed two retrieval strategies, Image-to-Image (I2I) and Text-to-Image (T2I), a direct comparison with the I2I strategy is incompatible with our setting due to its reliance on labeled training images as seed images. Instead, we compared our method against the T2I strategy. Unlike the T2I approach in [1], which retrieves a **fixed** set of external images, RA-TTA **adaptively** retrieves a **customized** set of external images for each test image, achieving superior performance, as shown in Table R1.
>
> Table R1: Performance for the T2I approach of [1] and the proposed RA-TTA. We report the top-1 accuracy (%) on the FGVC aircraft, Stanford cars, and RESISC45 datasets. The best results are in **bold**. The "Avg." column indicates the average accuracy of 3 datasets.
> | **Method** |   FGVC aircraft   | Stanford cars | RESISC45  | Avg. (3) |
> |------------|:-----------------:|:-------------:|:---------:|:---------:|
> | **T2I-adapter [1]**|      29.34        |     66.57     |   64.42   | 53.44 |
> | **RA-TTA** |    **31.51**      |   **69.43**   | **65.14** | **55.36**|
>
> ```Q2(W3): Why is optimal transport used, and what is its purpose?```
>
> **Optimal Transport (OT)** is a mathematical framework for calculating the distance between two distributions, typically represented by sampled data points. A key advantage of the OT framework is its **flexibility to incorporate a user-defined disparity function** to measure the distance between two distributions. When calculating this distance, OT requires a disparity matrix representing pairwise disparities between samples from the two distributions.
>
> In RA-TTA, a novel disparity function called **semantic gap** is introduced to calculate the semantic disparity between a test image and a retrieved image. Moreover, we adopt two techniques to enhance the reliability and robustness of the adaptation procedure:
> - We reuse the augmented images of the test image, previously employed in description-based retrieval. As a result, we have the test image set $\mathcal{A}$ (comprising the test image and its augmented images) and the retrieved image set $\mathcal{R}_{\mathcal{G}_c}$ that is related to the selected features of class $c$.
> - We represent the importance of each image within these sets using alignment scores because the importance of individual images might vary. This importance is normalized, allowing the two sets to be interpreted as probability distributions.
>
> Thus, calculating the distance between the test image and the retrieved images is transformed into calculating the distance between two distributions using our novel disparity function, **semantic gap**. Therefore, we adopt the optimal transport framework in this context.
>
> To enhance the clarity of the paper, we have included the preliminary of optimal transport in **Appendix A.1** of the revised manuscript and the discussion of optimal transport with RA-TTA in **Appendix A.2** of the revised manuscript.

---

> ### Author Response · Authors · 2024-11-22
> **Author's Response to Reviewer aU8U (2/2)**
>
> ```Q3: Figure 3 is not very clear in presenting the proposed method, specifically in the prototype-based retrieval part.```
>
> Thanks for your comments, which have helped improve the clarity of our paper. We conjecture that the absence of a legend for icons in the prototype-based retrieval part may have led to obscurity. Accordingly, we have edited and made overall revisions to Figure 3, including the addition of a legend. We have added the edited Figure 3 to the revised manuscript.
>
> ```W2: The proposed method achieves inferior performance than TPT [2] and RLCF [3] in IN-V2 and IN-A. The authors should analyze the reasons behind this phenomenon.```
>
> Thanks for letting us know your concern regarding performance. We speculate that the out-of-distribution property of the datasets might compromise the performance of the proposed RA-TTA for IN-V2 and IN-A. Specifically, IN-V2 and IN-A include samples that exist in the class boundary and require rare features rather than generalizable ones. However, the text descriptions of RA-TTA represent the general features of each class. Given that RA-TTA searches the features of a test image amidst the text descriptions, the absence of rare features in the descriptions renders it impossible to identify the features of a test image and to retrieve appropriate external images, resulting in performance decline. Nonetheless, RA-TTA attained the second-best performance, with a negligible difference to the best performance, i.e., 0.16% and 1.31%, respectively.
>
> In this study, we generated candidate text descriptions offline in accordance with CuPL [4]. At test time, we adaptively identified the relevant descriptions among the generated descriptions without additional generation. That is, the candidate text descriptions are fixed during test time. Thus, a **possible solution** is to adaptively generate text descriptions on the fly. We leave this exploration as an intriguing avenue for future work.
>
> Once again, we appreciate the reviewer highlighting valuable discussion points in our study. We hope our response has clarified the contents of the paper, and we are more than willing to engage in further discussion regarding any aspects that remain unclear.
>
> -----------------------------------------------------------------------
> [1] "Understanding Retrieval-Augmented Task Adaptation for Vision-Language Models.", In ICML, 2024
> [2] "SuS-X- Training-Free Name-Only Transfer of Vision-Language Models.", In ICCV, 2023
> [3] "Neural Priming for Sample-Efficient Adaptation.", In NeurIPS, 2023
> [4] "What Does a Platypus Look Like? Generating Customized Prompts for Zero-shot Image Classification.", In ICCV, 2023

---

> ### Author Response · Authors · 2024-11-26
> **Kindly Follow-Up: Reviewer aU8U**
>
> Dear Reviewer: aU8U,
>
> Thank you for your review and valuable feedback, which have greatly contributed to enhancing the quality of our paper. This is a kind reminder that we only have one and a half days (about 36 hours) before the author/reviewer discussion period ends. We have submitted a response that addresses your questions, providing clarification on novelty and the use of optimal transport. Additionally, we revised Figure 3. We hope that our response has provided sufficient clarification. Please let us know if you have any further inquiries or require additional information.
>
> Best regards,
> Authors.

---

> > ### Comment · Reviewer_aU8U · 2024-11-26
> > **Reply to Authors' Rebuttal**
> >
> > Thank you for the authors' response. However, I still believe that the entire method is based on [1], and simply improving its applicability is not enough to qualify it as a groundbreaking new contribution. So I keep my score.

---

> ### Author Response · Authors · 2024-11-26
> **Further Response to Reviewer aU8U: Clear Distinction from [1]**
>
> Thank you for your quick reply. We would like to clarify and discuss the details of our novelty.
>
> The retrieval-and-adaptation pipeline adopted in [1] has served as a common practice for retrieval-based adaptation approaches [1, 2, 3], and it is therefore **acceptable** to adopt this pipeline. It should be noted that [1] did not propose the pipeline for the first time; rather, [1] also simply adopted the pipeline following the common practice. The point of the retrieval-based adaptation is to **enhance** the pipeline. From this perspective, we respectfully contend that our work has clear novelty (far from improving applicability) compared to [1] in two aspects.
>
> **`Offline vs. Online:`** In contrast to all the previous studies that enhanced the pipeline in the **offline** scenario (i.e., before test time) [1, 2, 3], we are the **first** to study retrieval-based **test-time** adaptation, aiming to adapt VLMs to a test image provided **online**, which is effective for distribution shifts in the real world.
>
> **`Coarse-Grained vs. Fine-Grained:`** In terms of algorithmic detail, the previous approaches [1, 2, 3] used a **coarse-grained retrieval** that uses class names, for which the retrieved images can show irrespective features to a test image. On the other hand, we have newly developed a **fine-grained** approach (i.e., description-based retrieval) based on the **fine-grained features** of a test image, retrieving more relevant images that have targeted features. For this, an emerging challenge of identifying the features of a test image should be solved. This challenge is unique to us and **cannot be addressed by previous approaches**. We address this challenge by fully leveraging the cross-modal understanding of VLMs.
>
> We appreciate your concerns and welcome further discussions to refine and improve our work.
>
> --------------------------
> [1] "Understanding Retrieval-Augmented Task Adaptation for Vision-Language Models.", In ICML, 2024.
> [2] "SuS-X- Training-Free Name-Only Transfer of Vision-Language Models.", In ICCV, 2023.
> [3] "Neural Priming for Sample-Efficient Adaptation.", In NeurIPS, 2023.

---

> ### Author Response · Authors · 2024-11-29
> **Kindly Follow-Up: Reviewer aU8U**
>
> Dear Reviewer: aU8U
>
> Thank you again for your time and efforts in reviewing our paper.
>
> We have thoroughly answered your questions about **our novelty compared to [1]**.
>
> We would appreciate the opportunity to address any additional concerns you may have before the discussion phase ends.
>
> Best wishes,
> Authors.

---

### Official Review · Reviewer_gDjz · 2024-11-02

**Soundness:** 2
**Presentation:** 2
**Contribution:** 2
**Rating:** 5
**Confidence:** 5

**Summary:**

The paper proposes a novel retrieval-augmented test-time adaptation (RA-TTA) for improving VLM generalization on downstream datasets at test time. RA-TTA exploits additional visual information through retrieval of web-scale images to regularize VLM predictions at test time. Extensive experiments over various datasets show the effectiveness of the proposed method.

**Strengths:**

The introduction retrieval augmentation into TTA problem is an innovative approach. Also, the proposed description-based retrieval technique aligns well with the multi-modal capabilities of VLMs.

The paper presents extensive experimental evaluations over 17 diverse datasets. The authors also conduct detailed ablation studies and sensitivity analyses.

The paper is well-organized and easy to follow.

**Weaknesses:**

The paper lacks some necessary discussions and analyses.

(1) In the proposed method, different databases are sampled from large-scale datasets for each downstream dataset respectively. This may introduce additional computational requirements when adapting to different datasets. Will directly retrieving from the large-scale datasets like LAION2B help mitigate this issue? In addition, the computation overhead or inference time compared with TPT should be provided for examining the efficiency of the proposed method.

(2) The retrieval process highly depends on the LLMs for generating detailed text descriptions. The authors are suggested to discuss how different LLMs might affect description quality and, consequently, the final adaptation performance.

(3) How do the authors ensure that the retrieved images contain the expected features to augment the test image? For example, in the examples provided in the introduction, if the test image depicts the front and side of a vehicle, the retrieved images should ideally contain related information, such as headlights.

Some recent and related studies are missing from the section Related work. The author should consider discussing the difference with these works [a,b,c,d].
[a] Efficient Test-Time Adaptation of Vision-Language Models
[b] Dual memory networks: A versatile adaptation approach for vision-language models
[c] SwapPrompt: Test-Time Prompt Adaptation for Vision-Language Models
[d] Historical Test-time Prompt Tuning for Vision Foundation Models

**Questions:**

as shown in Weaknesses

---

> ### Author Response · Authors · 2024-11-22
> **Author's Response to Reviewer gDjz (1/2)**
>
> We deeply appreciate the reviewers’ valuable comments and reasonable concerns. We hope that they can be resolved through our clarifications in this rebuttal.
>
> ```Q1-1: Will directly retrieving from the large-scale datasets like LAION2B help mitigate this issue?```
>
> Yes, directly using LAION2B can mitigate the effort required to construct tailored databases for each downstream dataset. However, in our case, constructing tailored databases can be conducted efficiently using SQLite's full text search (FTS). Moreover, constructing tailored databases is a more practical and effective approach than utilizing all images from web-scale datasets, where the characteristics of downstream tasks, such as class names or domains, are known in advance.
>
> ```Q1-2: In addition, the computation overhead or inference time compared with TPT should be provided for examining the efficiency of the proposed method.```
>
> This is a good point. In the context of test-time adaptation, the efficiency of adaptation at test time is indeed a critical factor to consider. In contrast to previous TTA methods, RA-TTA needs to retrieve proper external images for each test image, potentially posing a significant bottleneck. However, the efficiency of RA-TTA is **comparable** to the representative test-time adaptation method, TPT, as shown in Table R1, where we compared the GPU inference time per sample (i.e., s/sample) for TPT [1] and RA-TTA. This is because FAISS, an advanced and efficient search engine typically used for RAG-based approaches, enables **fast nearest-neighbor search**. Furthermore, our description-based adaptation does not require backpropagation (i.e., training-free), which also contributes to the efficient inference of RA-TTA.
>
> We fully agree with the reviewer on the importance of the efficiency analysis, and we have added it to **Appendix D.4** of the revised manuscript.
>
> Table R1: Inference time for TPT [1] and the proposed RA-TTA. We report the average GPU inference time per sample (s/sample) measured on a single RTX4090 for the FGVC aircraft, Stanford cars, and RESISC45 datasets. The best results are in **bold**. The "Avg." column indicates the average inference time of 3 datasets.
> | **Method** |   FGVC aircraft (s/sample)  | Stanford cars (s/sample)  |  RESISC45 (s/sample) | Avg. (3) |
> |------------|:----------------:|:---------------:|:----------:|:----------:|
> | **TPT** [1]    |     **0.103**    |    0.155     |  **0.95** | 0.118|
> | **RA-TTA**     |     0.113        |    **0.117**    |    0.121   | **0.117**|
>
>
> ```Q2: The authors are suggested to discuss how different LLMs might affect description quality and, consequently, the final adaptation performance.```
>
> We appreciate the reviewer for providing a valuable discussion point. We conducted additional experiments using text descriptions generated by Claude3-Sonnet for this analysis.
> - We also compared the performance of CuPL [2] to investigate the quality of text descriptions because CuPL can achieve better performance with improved descriptions. As shown in the first two rows of Table R2, CuPL with the descriptions of GPT works better than CuPL with those of Claude, implying that the text descriptions generated by GPT are better than those of Claude. Thus, we can conclude that different LLMs can affect description quality even with the same generation prompts.
> - Also, the last two rows of Table R2 show that RA-TTA with the descriptions of GPT outperformed RA-TTA with those of Claude. This indicates that the better descriptions facilitate the adaptation of RA-TTA more effectively.
> - Notably, RA-TTA with the text descriptions of Claude outperformed the performance of corresponding CuPL (i.e., CuPL+Claude3 vs. RA-TTA+Claude3), demonstrating that RA-TTA effectively leverages external knowledge, even when the quality of the description is not very good.
>
> We have added this analysis to **Appendix D.5** of the revised manuscript.
>
> Table R2: Performance when using text descriptions generated by ChatGPT and Claude3. We report the top-1 accuracy (%) on the FGVC aircraft, Stanford cars, and RESISC45 datasets. The employed LLMs are indicated with the method name. The "Avg." column indicates the average accuracy of 3 datasets.
> | **Method+LLM** | FGVC aircraft    | Stanford cars | RESISC45 | Avg(3)     |
> |----------- |:------------------:|:---------------:|:---------:|:---------:|
> | **CuPL+Claude3**    |  28.05  |    66.38      |   57.33  |   50.59 |
> | **CuPL+ChatGPT**    |  28.89  |    66.78      |   63.98  |   53.22 |
> | **RA-TTA+Claude3**  |  30.93  |    68.82      |   61.92  |   53.80 |
> | **RA-TTA+ChatGPT**  |  **31.51**  |    **69.43**      |   **65.14**  | **55.36** |

---

> ### Author Response · Authors · 2024-11-22
> **Author's Response to Reviewer gDjz (2/2)**
>
> ```Q3: How do the authors ensure that the retrieved images contain the expected features to augment the test image?```
>
> The proposed RA-TTA fully leverages the cross-modal understanding of a VLM to associate an image with the corresponding textual descriptions. Recent VLM-based works [3, 4, 5, 6, 7, 8, 9] also have adopted this cross-modal association. However, we acknowledge that the cross-modal association may occasionally retrieve irrelevant images that lack the expected features. Recent studies have enhanced the association capability of a VLM by transferring the knowledge of LLMs [10, 11, 12] or the diffusion model [13] or introducing novel training strategies [14]. Leveraging the enhanced association capability of these studies, we can improve the relevance of retrieved images.
>
> Additionally, we utilized the **prototypes** of selected features in the description-based retrieval to enhance the relevance of retrieved images. By incorporating diverse features with different semantics, the prototypes represent the target image more comprehensively, thereby increasing the likelihood of retrieving relevant images.
>
> ```Q4: The author should consider discussing the difference with these works.```
>
> Thanks for your thoughtful feedback, which has helped us better position our paper compared to the related works. Both the related works [15, 16, 17, 18] and the proposed RA-TTA utilize additional knowledge for adaptation. The key distinction lies in **the source of knowledge**: the related works rely on **historical test samples** as additional knowledge for adaptation, whereas RA-TTA leverages **external samples** from an **external** database. We hypothesize that these two sources of knowledge (i.e., historical and external) are complementary to each other, and integrating both knowledge could be a promising direction for enhancing test-time adaptation. We leave this exploration as an intriguing avenue for future work.
>
> The discussion regarding these related works has been incorporated into **Appendix A.1** of the revised manuscript.
>
> We are once again grateful for the reviewer's valuable discussion points in our study. We are eager to engage in further discussion regarding any aspects that remain unclear.
>
> -----------------------------------------------------------
> [1] "Test-Time Prompt Tuning for Zero-Shot Generalization in Vision-Language Models.", In NeurIPS, 2022
> [2] "What Does a Platypus Look Like? Generating Customized Prompts for Zero-shot Image Classification.", In ICCV, 2023
> [3] "CLIPScore: A Reference-free Evaluation Metric for Image Captioning.", In EMNLP, 2021
> [4] "Scaling Up GANs for Text-to-Image Synthesis.", In CVPR, 2023
> [5] "Visual Classification via Description from Large Language Models.", In ICLR, 2023
> [6] "CLIP-Dissect: Automatic Description of Neuron Representations in Deep Vision Networks.", In ICLR, 2023
> [7] "Label-free Concept Bottleneck Models.", In ICLR, 2023
> [8] "What Does a Platypus Look Like? Generating Customized Prompts for Zero-shot Image Classification.", In ICCV, 2023
> [9] "Discovering and Mitigating Visual Biases through Keyword Explanation.", In CVPR, 2024
> [10] "MATE: Meet At The Embedding - Connecting Images with Long Texts.", In EMNLP findings, 2024
> [11] "InternVL: Scaling up Vision Foundation Models and Aligning for Generic Visual-Linguistic Tasks,.", In CVPR, 2024
> [12] "LLM2CLIP: Powerful Language Model Unlocking Richer Visual Representations.", arXiv preprint arXiv:2411.04997, 2024
> [13] "Diffusion Feedback Helps CLIP See Better.", arXiv preprint arXiv:2407.20171, 2024
> [14] "Long-CLIP: Unlocking the Long-Text Capability of CLIP.", In ECCV, 2024
> [15] "Efficient Test-Time Adaptation of Vision-Language Models.", In CVPR, 2024
> [16] "Dual Memory Networks: A Versatile Adaptation Approach for Vision-Language Models.", In CVPR, 2024
> [17] "SwapPrompt: Test-Time Prompt Adaptation for Vision-Language Models.", In NeurIPS, 2023
> [18] "Historical Test-time Prompt Tuning for Vision Foundation Models.", In NeurIPS, 2024

---

> ### Author Response · Authors · 2024-11-26
> **Kindly Follow-Up: Reviewer gDjz**
>
> Dear Reviewer: gDjz,
>
> Thank you for your review and valuable feedback, which have greatly contributed to enhancing the quality of our paper. This is a kind reminder that we only have one and a half days (about 36 hours) before the author/reviewer discussion period ends. We submitted an answer addressing your questions, including the computation overhead and the usage of a different LLM. We hope that our response has provided sufficient clarification. Please let us know if you have any further inquiries or require additional information.
>
> Best regards,
> Authors.

---

> ### Author Response · Authors · 2024-11-29
> **Kindly Follow-Up: Reviewer gDjz**
>
> Dear Reviewer: gDjz
>
> Thank you again for your time and efforts in reviewing our paper.
> We have answered your questions, provided new experiments as required, and updated our manuscript.
>
> We would appreciate the opportunity to address any additional concerns you may have before the discussion phase ends.
>
> Thank you very much.
> Best wishes,
> Authors.

---

### Official Review · Reviewer_UYSy · 2024-11-03

**Soundness:** 3
**Presentation:** 4
**Contribution:** 3
**Rating:** 6
**Confidence:** 5

**Summary:**

This paper introduces the success of RAG into VLM for test-time adaptation, achieving performance improvements. Extensive experiments demonstrate the effectiveness of the method.

**Strengths:**

1. This paper is well-written, and the framework is clear and easy to follow. Additionally, the provided code enhances reproducibility.
2. Introducing RAG into VLM for test-time adaptation is a reasonable and meaningful idea.
3. The experiments in this paper are thorough, with detailed descriptions. The method surpasses others on multiple datasets.

**Weaknesses:**

1. More training-free methods should be discussed and compared. In VLM-based transfer learning, training-free adapters [1] have significantly improved with a small number of image samples, as in CaF [2] with fine-grained descriptions based on classes. The proposed method is more complex, so I am curious about its performance advantage compared to these easy but efficient methods.

2. This method involves generating descriptors for each test sample with a GPT-based generator, then self-filtering with the image itself, retrieving similar samples, and finally drawing conclusions. This is a relatively lengthy pipeline. How does its efficiency compare to other methods? Additionally, GPT usage is not free, though open-source models could be a workaround. Could this pipeline be simplified, perhaps by using improved prompts to avoid generating imaginary content or by creating visual descriptors to replace the image-to-description selection?

3. Could more visualization results be provided, showing the generated descriptors -> filtered irrelevant descriptors -> refined inference results, similar to the presentation in Figures 3 and 4 of DVDet [3]?

References:

[1] Tip-Adapter: Training-free CLIP-Adapter for Better Vision-Language Modeling
[2] Visual Classification via Description from Large Language Models
[3] LLMs Meet VLMs: Boost Open Vocabulary Object Detection with Fine-Grained Descriptors

**Questions:**

Please see the weakness.

---

> ### Author Response · Authors · 2024-11-22
> **Author's Response to Reviewer UYsy (1/2)**
>
> We sincerely thank the reviewer for your insightful comments, which have helped us improve the paper quality significantly. Below, we are more than willing to clarify the reviewer’s concerns one by one.
>
> ```Q1: More training-free methods should be discussed and compared.```
>
> The goal of the proposed RA-TTA is to leverage **external knowledge** when transferring a VLM at test time. For this purpose, we proposed RA-TTA, an end-to-end pipeline that includes a novel description-based retrieval and adaptation. However, the training-free methods [1, 2] focused on adapting a VLM without backpropagation, which is **orthogonal** to leveraging external knowledge. Consequently, RA-TTA incorporated external knowledge into adaptation at test time that is overlooked by the training-free methods, achieving performance improvement.
>
> Additionally, to compare the performance of RA-TTA with that of the training-free methods, including Tip-Adapter [1] and CaFo [2], a retrieval approach is needed to enable these training-free methods to leverage external knowledge. We adopted a simple retrieval approach [3, 4] and implemented Tip-Adapter and CaFo on top of it. As shown in Table R1, RA-TTA outperformed the variants because RA-TTA is designed to effectively leverage external knowledge at test time, whereas the training-free variants do not account for using external knowledge.
>
> The discussion regarding training-free methods has been incorporated into **Appendix A.1** of the revised manuscript.
>
> Table R1: Performance for the retrieval-based variants of training-free methods and the proposed RA-TTA. We report the top-1 accuracy (%) on the FGVC aircraft, Stanford cars, and RESISC45 datasets. The best results are in **bold**. The "Avg." column indicates the average accuracy of 3 datasets.
> | **Method** | FGVC aircraft     | Stanford cars | RESISC45 | Avg. (3) |
> |------------|:-------------------:|:---------------:|:----------:|:----------:|
> | **Tip-Adapter [1]+Retrieval**  |  29.34    | 66.57 |   64.42    | 53.44 |
> | **CaFo [2]+Retrieval**         |  29.40    | 67.19 |   64.74    | 53.78 |
> | **RA-TTA**                     | **31.51** | **69.43** | **65.14** | **55.36** |
>
> ```Q2-1: How does its efficiency compare to other methods?```
>
> This is a good point. In the context of test-time adaptation, the efficiency of adaptation at test time is indeed a critical factor to consider. In contrast to previous TTA methods, RA-TTA needs to retrieve proper external images for each test image, potentially posing a significant bottleneck. However, the efficiency of RA-TTA is **comparable** to the representative test-time adaptation method, TPT, as shown in Table R2, where we compared the GPU inference time per sample (i.e., s/sample) for TPT [5] and RA-TTA. This is because FAISS, an advanced and efficient search engine typically used for RAG-based approaches, enables **fast nearest-neighbor search**. Furthermore, our description-based adaptation does not require backpropagation (i.e., training-free), which also contributes to the efficient inference of RA-TTA.
>
> Thank you again for your insightful comments. We have added this efficiency analysis in **Appendix D.4** of the revised manuscript.
>
> Table R2: Inference time for TPT [5] and the proposed RA-TTA. We report the average GPU inference time per sample (s/sample) measured on a single RTX4090 for the FGVC aircraft, Stanford cars, and RESISC45 datasets. The "Avg." column indicates the average inference time of 3 datasets.
> | **Method** |   FGVC aircraft (s/sample)  | Stanford cars (s/sample)  |  RESISC45 (s/sample) | Avg. (3) |
> |------------|:----------------:|:---------------:|:----------:|:----------:|
> | **TPT** [5]    |     **0.103**    |    0.155     |  **0.95** | 0.118|
> | **RA-TTA**     |     0.113        |    **0.117**    |    0.121   | **0.117**|
>
> ```Q2-2: GPT usage is not free.```
>
> The generation of text descriptions relies on GPT3.5-turbo, which does incur some costs. However, this cost is **negligible** because our query to GPT is relatively simple, such as "Describe what a 2012 Porsche Panamera Sedan, a type of car, looks like." For instance, generating text descriptions for the Stanford cars dataset that includes 196 classes incurs an expense of only $2.73.
>
> Furthermore, these text descriptions are generated **offline**, which is conducted **only once** prior to test time; thus, no additional costs are incurred during inference at test time.

---

> ### Author Response · Authors · 2024-11-22
> **Author's Response to Reviewer UYsy (2/2)**
>
> ```Q2-3: Could this pipeline be simplified, perhaps by using improved prompts to avoid generating imaginary content or by creating visual descriptors to replace the image-to-description selection?```
>
> Thank you very much for your insightful ideas. We can simplify the pipeline using your ideas.
>
> - First, given that multiple prompts and queries are required to generate semantically diverse descriptions, we can simplify the description generation procedure with improved prompts that can elicit text descriptions with diverse semantics with a few iterations.
> - Second, we can streamline the pipeline by replacing the image-to-description selection with visual descriptors generated by an image generator like Stable Diffusion.
>
> Though these ideas are very interesting topics, it seems to be beyond the scope of this paper. We leave these topics for potential future work.
>
> ```Q3: Could more visualization results be provided? ```
>
> Thanks for your suggestion, which improves the elucidation of our paper. We have added the visualization results in **Appendix D.7** of the revised manuscript.
>
> Once again, we appreciate the reviewer highlighting valuable discussion points in our study. We hope our response has clarified the contents of the paper, and we are more than willing to engage in further discussion regarding any aspects that remain unclear.
>
> -----------------------------------------
> [1] "Tip-Adapter: Training-Free CLIP-Adapter for Better Vision-Language Modeling.", In ECCV, 2022
> [2] "Prompt, Generate, then Cache: Cascade of Foundation Models Makes Strong Few-Shot Learners.", In CVPR, 2023
> [3] "SuS-X: Training-Free Name-Only Transfer of Vision-Language Models.", In ICCV, 2023
> [4] "Understanding Retrieval-Augmented Task Adaptation for Vision-Language Models.", In ICML, 2024
> [5] "Test-Time Prompt Tuning for Zero-Shot Generalization in Vision-Language Models.", In NeurIPS, 2022

---

> > ### Comment · Reviewer_UYSy · 2024-11-24
> > **Reply to Authors' Rebuttal**
> >
> > The authors address most of my concerns. I will maintain the positive scores.

---

> > > ### Author Response · Authors · 2024-11-25
> > >
> > > We sincerely appreciate your acknowledgment of our response. Moreover, we are truly grateful for the time and consideration you have invested in reviewing our manuscript.
> > >
> > > If you have any further questions or concerns, please let us know. We are more than willing to address any additional questions or concerns the reviewer might have.
> > >
> > > With warmest regards,
> > > Authors.

---

### Author Response · Authors · 2024-11-22
**Response Summary to All Reviewers**

Dear Reviewers,
We would like to express our sincere gratitude to all the reviewers for their constructive feedback on our manuscript. Your insights have been valuable in improving the quality and clarity of our work. We also sincerely thank you for acknowledging the novelty of introducing retrieval augmentation into test-time adaptation of VLMs (UYsy, gDjz, DF2M), the logical structure of our presentation and illustration (UYsy, gDjz, aU8U), and the comprehensive evaluation (UYsy, gDjz, aU8U, DF2M).

Following the reviewers' comments, we have endeavored to address the concerns raised:
- **Additional analyses**: We have provided analyses, including the efficiency analysis (UYsy, gDjz, DF2M), the usage of different LLMs (gDjz, DF2M), and the adaptation performance with respect to the database size (DF2M).
- **Discussion about related works**: We have enriched our discussion and comparison of related works to better position our contributions within the existing body of research (UYsy, gDjz, aU8U).
- **Clarity improvements**: We have improved the clarity of our manuscript with more visualization results (UYsy), an elaborate preliminary (aU8U), and a revised figure (aU8U).

We have uploaded our revised manuscript. We highlighted the revised parts in blue. The updates include:
- **Revision of Figure 3**
- **Extended Related Works** in Appendix A.1
- **Preliminary** in Appendix A.1 and A.2
- **Efficiency Analysis** in Appendix D.4
- **The Usage of Different LLMs** in Appendix D.5
- **Database Size Analysis** in Appendix D.6
- **More Visualization Results** in Appendix D.7

Once again, thank you for your time and effort in reviewing our work. We look forward to your continued feedback.

Best regards,

Authors

---

### Meta-Review · Area_Chair_1MDT · 2024-12-20

**Metareview:**

(a) Scientific Claims and Findings

The paper proposes a retrieval-augmented test-time adaptation (RA-TTA) method for vision-language models (VLMs) to enhance their generalization on downstream datasets. By incorporating external visual information through retrieval of web-scale images with fine-grained text descriptions, RA-TTA aims to improve VLM predictions at test time. Reviewers note that the method is evaluated extensively across 17 datasets, demonstrating its effectiveness in surpassing existing test-time adaptation methods.

(b) Strengths

Reviewer UYSy highlights the paper's clarity, thorough experiments, and the meaningful introduction of RAG into VLM for test-time adaptation. gDjz appreciates the approach of retrieval augmentation and the experimental evaluations. aU8U commends the investigation into test-time adaptation for VLMs and the state-of-the-art results achieved on multiple benchmarks. DF2M notes the novelty of using retrieval augmentation in TTA and the comprehensive analysis provided through experiments.

(c) Weaknesses

The reviewers point out several weaknesses. UYSy suggests that more training-free methods should be discussed and questions the efficiency of the proposed pipeline. gDjz raises concerns about the computational requirements and the dependency on LLMs for generating text descriptions. aU8U criticizes the lack of innovation, noting similarities with existing methods, and questions the clarity of certain operations. DF2M highlights the reliance on LLMs and the potential computational overhead compared to prior studies, suggesting a need for analysis on database size and semantic diversity in text descriptions.

(d) Decision Reasons

On balance, AC agrees with positive points raised by the reviewers which outweigh the negative points which are mainly focusing on existing work (novelty concern). The authors are strongly encouraged to include the additional reviewer recommendations, experiments from rebuttal and clarifications in the camera-ready version.

**Additional Comments On Reviewer Discussion:**

During the rebuttal period, the authors addressed several concerns raised by the reviewers, leading to some adjustments in their evaluations, though not all reviewers changed their scores.

Reviewer UYSy expressed satisfaction with the authors' responses, noting that most of their concerns were addressed. As a result, they decided to maintain their positive scores, indicating continued support for the paper's acceptance.

Reviewer aU8U acknowledged the authors' response but remained unconvinced about the novelty of the contribution. They reiterated that the method closely resembles existing work [1] and that simply improving its applicability does not constitute a groundbreaking new contribution. aU8U' main concern is about similarity to an existing work, which the authors added in their revision. The authors compare to this work and argue that this work is one among several similar ones. The reviewer maintains their slightly negative score after rebuttal.

Reviewer DF2M appreciated the authors' efforts in addressing their comments and concerns. After reviewing the responses, they acknowledged the improvements made but felt that their initial evaluation and rating were still appropriate. Therefore, they decided to maintain their original rating of weak accept.

In weighing these points for the final decision, the primary considerations were the positive feedback from reviewers UYSy and DF2M, who recognized the paper's contributions and improvements. While reviewer aU8U raised concerns about the novelty, the overall assessment from the other reviewers, combined with the experimental validation, supported the decision to accept the paper. The authors' ability to address most concerns during the rebuttal period strengthened the case for acceptance.

---

### Decision · Program_Chairs · 2025-01-22

Accept (Poster)